# The intercentriolar fibers function as docking sites of centriolar satellites for cilia assembly

Sungjin Ryu[1]*, Donghee Ko[1]*, Byungho Shin[1], and Kunsoo Rhee[1]

**Two mother centrioles in an animal cell are linked by intercentriolar fibers that have CROCC/rootletin as their main building block. Here, we investigated the regulatory role of intercentriolar/rootlet fibers in cilia assembly. The cilia formation rates were significantly reduced in the *CEP250/C-NAP1* and *CROCC/rootletin* knockout (KO) cells, irrespective of the departure of the young mother centrioles from the basal bodies. In addition, centriolar satellites were dispersed throughout the cytoplasm in the *CEP250* and *CROCC* KO cells. We observed that PCM1 directly binds to CROCC. Their interaction is critical not only for the accumulation of centriolar satellites near the centrosomes/basal bodies but also for cilia formation. Finally, we observed that the centriolar satellite proteins are localized at the intercentriolar/rootlet fibers in the kidney epithelial cells. Based on these findings, we propose that the intercentriolar/rootlet fibers function as docking sites for centriolar satellites near the centrosomes/basal bodies and facilitate the cilia assembly process.**

## Introduction

The centrosome, which is the major microtubule organizing center in animal cells, consists of a (pair of) centriole(s) and a surrounding protein matrix called pericentriolar material. Centriole assembly and segregation are tightly linked with the cell cycle. During the S phase of the cell cycle, a daughter centriole assembles next to a mother centriole and remains associated until the cell exits the mitosis. During mitosis, a centrosome with a pair of centrioles functions as a spindle pole to pull a set of chromosomes into daughter cells. At the end of mitosis, the daughter centriole separates from the mother centriole and becomes a young mother centriole. As a result, both young and old mother centrioles always exist in a single cell. During interphase, two mother centrioles are linked by intercentriolar fibers. When the cell approaches mitosis, the intercentriolar fibers dissolve to allow two centrosomes to become spindle poles.

CROCC/rootletin and CEP68 are known components of the intercentriolar fiber (Bahe et al., 2005; Graser et al., 2007). Additional proteins, such as LRRC45, centlein, CCDC102B, and CEP44, are also present in the fiber (He et al., 2013; Fang et al., 2014; Xia et al., 2018; Hossain et al., 2020). CROCC is the main building block of intercentriolar/rootlet fibers and forms a parallel homodimer as a basic unit (Ko et al., 2020). The intercentriolar/rootlet fibers have indented lines that are 75 nm apart, suggesting that CROCC dimers are ordered in a staggered manner (Yang et al., 2002; Vlijm et al., 2018; Ko et al., 2020). The fibers are anchored to the proximal ends of the mother

centrioles via CEP250/C-NAP1 (Fry et al., 1998). Based on microscopic analysis, the intercentriolar/rootlet fibers form flexible, dynamic, and interdigitating networks (Mahen, 2018; Vlijm et al., 2018).

Centriolar satellites are 70–100-nm nonmembranous granules that undergo cell cycle–dependent assembly and disassembly (Hori and Toda, 2017). Centriolar satellites move along microtubules toward the centrosome in a dynein-dependent manner (Dammermann and Merdes, 2002; Kubo and Tsukita, 2003). PCM1 is a key scaffold protein (Kubo et al., 1999), and over 65 proteins are known to colocalize with PCM1 at centriolar satellites (Prosser and Pelletier, 2020). Owing to recent proteome and interactome analyses, the number of satellite-associated proteins has expanded to hundreds, revealing the diverse composition of centriolar satellites (Gheiratmand et al., 2019; Quarantotti et al., 2019). The most well-known function of centriolar satellites is protein trafficking to the centrosome. Mutations in several genes that encode satellite proteins are linked to ciliopathies, revealing the involvement of centriolar satellites in cilia assembly and maintenance (Nachury et al., 2007; Kim et al., 2008a). Loss of PCM1 does not affect cell cycle progression or centriole duplication, but instead induces marked defects in cilia formation in certain cell types (Wang et al., 2016; Odabasi et al., 2019).

Primary cilia protrude from the cell surface and function as signaling antennae in many mammalian cells. Primary cilia originate from the old mother centrioles, which possess distal

[1]Department of Biological Sciences, Seoul National University, Seoul, South Korea.

*S. Ryu and D. Ko contributed equally to this paper. Correspondence to Kunsoo Rhee: rheek@snu.ac.kr.



and subdistal appendages, and are formed under a tightly regulated, multistep process (Breslow and Holland, 2019). The nine doublet microtubules of the ciliary axoneme are formed through the elongation of the A and B tubules of the old mother centriole. The centriolar distal appendages form an interface that connects the centriole to the nascent ciliary membrane and anchors it to the cell surface when a mature cilium assembles (Reiter et al., 2012). Although the ciliary membrane is continuous with the plasma membrane, the cilia maintain a unique complement of biomolecules through the combined action of dedicated trafficking machinery and diffusional barriers at the cilium base (Nachury et al., 2010). The ciliary components must be actively transported for cilia assembly and maintenance (Kumar and Reiter, 2021). As a result, the old mother centriole must play a dual role, acting both as a basal body that anchors a primary cilium and as a centrosome that organizes microtubules (Breslow and Holland, 2019).

In addition to the structural importance of the intercentriolar/rootlet fibers for the stability of primary cilia, their regulatory implications were also proposed. For example, the *rootletin* mutant flies revealed behavioral defects associated with mechano- and chemosensation (Styczynska-Soczka and Jarman, 2015; Chen et al., 2015). In this study, we investigated another implication of the intercentriolar/rootlet fibers in the regulation of cilia assembly. We proposed that the intercentriolar/rootlet fibers may be a docking site for centriolar satellites facilitating cilia assembly.

## Results

### The intercentriolar/rootlet fibers are involved in cilia formation

We used RPE1 (retinal pigment epithelial 1) cells to investigate the specific roles of intercentriolar/rootlet fibers in cilia assembly. We initially observed that the CROCC/rootletin signal intensities in basal bodies were stronger than those in centrosomes (Fig. 1, A and B). The total cellular amount of the CROCC protein was unaffected during the cilia formation period (Fig. 1 C). Similar results were also observed in other cell lines, such as HK2 and IMCD3 (Fig. 1, D and E). Super-resolution microscopic analysis with the CROCC and CEP68 antibodies revealed that the intercentriolar/rootlet fibers became more extensive in basal bodies than in centrosomes (Fig. 1 F). However, the intensities of CEP250/C-NAP1 were more or less the same in both the basal bodies and centrosomes (Fig. 1 F). These results support the notion that the intercentriolar/rootlet fibers play a structurally supporting role in anchoring the cilia (Yang et al., 2002). In subsequent experiments, we found that the intercentriolar/rootlet fibers also have a regulatory role in cilia assembly.

We generated CROCC and CEP250 knockout (KO) cell lines using the CRISR/Cas9 method and confirmed the absence of the CROCC and CEP250 proteins with immunoblot and immunostaining analyses (Fig. 2 A; Fig. S1, A and B; and Fig. S2, A and B). In our experimental conditions with a cell density of 1.3 × $10^4$ cells/cm$^2$, over 60% of the RPE1 cells formed cilia in a serum-deprived medium (Fig. S3, A and B; Kim et al., 2012). However,

the cilia formation rates were significantly reduced to <40% in both the CEP250 and CROCC KO cell lines (Figs. 2, A and B; Fig. S1 C, and Fig. S2 C). These results indicate that the intercentriolar/rootlet linkers are required for the proper formation of cilia in RPE1 cells.

One can expect that the old mother centrioles/basal bodies and young mother centrioles would depart from each other after the removal of the intercentriolar linkers (Flanagan et al., 2017; Panic et al., 2015). In fact, the centriolar distances increased in the CEP250 KO cells but not in the CROCC KO cells (Fig. 2 C, Fig. S1 D, and Fig. S2 D). However, we observed that nocodazole, a microtubule destabilizer, augmented the centriole disjunction rates of all experimental groups, but more significantly in both the CEP250 and CROCC KO cells (Fig. 2 C and Fig. S1 D). These results are consistent with the fact that the centrosome pairs are linked with the microtubule network as well as with the intercentriolar fibers (Hata et al., 2019). The cilia formation rate decreased in the CEP250 KO cells, irrespective of the departure of young mother centrioles from the basal bodies (Fig. 2, D and E). These findings suggest that intercentriolar fibers are required for cilia assembly, while the presence of a young mother centriole near the basal body may not be a critical factor for the cilia assembly.

It is controversial whether cilia formation rates are reduced or not in the CEP250 KO cells (Panic et al., 2015; Mazo et al., 2016; Flanagan et al., 2017). First, we determined cilia formation rates in different culture conditions of RPE1 cells. We learned that cell density critically affects cilia formation rates so that 60% of RPE1 cells formed cilia in 1.3 × $10^4$ cells/cm$^2$, while about 80% of them formed cilia in more than 2.5 × $10^4$ cells/cm$^2$ (Fig. S3 B). In the high cell density condition, the cilia formation rate of the CEP250 KO cells increased to 75%, comparable with that of the wild-type cells (Fig. 2 F and Fig. S3 C). Perhaps other research groups might determine cilia formation rates of CEP250 KO cells in a high cell density condition. Interestingly, the cilia formation rate of CROCC KO cells appeared unaffected by cell density (Fig. S3 C).

Since we hypothesize that CROCC is involved in the efficient formation of cilia, we determined the centrosome levels of CROCC in the CEP250 KO cells. CROCC is placed at the centrosomes in about 30% of the CEP250 KO cells and this number went up to almost 50% in cells cultured in serum-deprived medium (Fig. 2, G and H). Furthermore, the proportion of the CEP250 KO cells with centrosome CROCC went up to 65% when the cells were cultured in high density (Fig. 2 H). When the CEP250 KO cells were cultured in a serum-deprived medium, most of those with cilia had CROCC at the centrosomes, while only half of those without cilia had the centrosome CROCC (Fig. 2 F). These results support the notion that the intercentriolar/rootlet fibers are important for the proper formation of cilia in the CEP250 KO cells.

### The intercentriolar/rootlet fibers are important for the accumulation of centriolar satellites in the centrosomes

The subcellular distribution of centriolar satellite proteins was determined in CEP250 and CROCC KO RPE1 cells. The major centriolar satellite proteins, such as PCM1, CEP290, OFD1,



**Figure 1. Expansion of the intercentriolar/rootlet fibers in cilia. (A)** RPE1 cells without and with cilia were subjected to coimmunostaining analysis with antibodies specific to CROCC (cyan) and acetylated tubulin (magenta). **(B)** Intensities of CROCC at the centrosomes and basal bodies were determined in cells without and with cilia, respectively. **(C)** RPE1 cells were cultured in serum-deprived medium for 0, 24, 48, and 72 h. The cells were subjected to immunoblot analysis with antibodies specific to CROCC and GAPDH. Average intensities of the CROCC-specific bands were indicated after the three repeated experiments. **(D)** HK2 and IMCD3 cells were cultured in a serum-deprived medium for 48 h. The cells were subjected to coimmunostaining analysis with antibodies specific to CROCC (cyan) and acetylated tubulin (magenta). **(E)** Intensities of CROCC at the centrosomes/basal bodies were determined in cells without and with cilia, respectively. **(F)** RPE1 cells were coimmunostained with antibodies specific to acetylated tubulin (magenta) along with CROCC (cyan), CEP68 (cyan), and CEP250 (cyan). **(A, D, and F)** Scale bars, 2 μm. **(B and E)** More than 15 cells per group were counted in three independent experiments. Within each box, the black center line represents the median value, the black box contains the interquartile range, and the black whiskers extend to the 10th and 90th percentiles. Statistical significance was determined using one-way ANOVA with Tukey's post hoc test (***, P < 0.001; n.s., not significant). Source data are available for this figure: SourceData F1.

CEP131, and CEP90, were found to be dispersed throughout the cytoplasm of *CEP250* and *CROCC* KO cells (Fig. 3, A and B). As a result, the intensities of the centriolar satellite proteins near the basal bodies/centrosomes were significantly reduced (Fig. 3, A and B). Nonetheless, the total amount of centriolar satellite proteins remained unchanged, irrespective of the *CEP250* and *CROCC* KO (Fig. 3 C). These results support the notion that

intercentriolar/rootlet fibers are important for the accumulation of centriolar satellites near the centrosome and cilia.

CEP72 is a PCM1-interacting, centriolar satellite protein that is involved in the delivery of ciliary proteins from satellites to cilia (Stowe et al., 2012). It is known that centriolar satellites are concentrated at the centrosomes in CEP72-depleted cells (Stowe et al., 2012; Conkar et al., 2019). In fact, we also observed that the



Figure 2. **Reduction of cilia assembly in the *CEP250* and *CROCC* KO cells. (A)** The *CEP250* and *CROCC* KO RPE1 cells were cultured in serum-deprived medium for 48 h and subjected to coimmunostaining analyses with antibodies specific to CEP250 (cyan), CROCC (cyan), and acetylated tubulin (magenta). **(B)** The number of cells with cilia was counted. **(C)** The number of cells with centriole disjunction (>2 μm) was counted after treatment of 20 μM nocodazole for 2 h. **(D)** The *CEP250* KO cells were cultured in serum-deprived medium for 48 h to induce cilia assembly and subjected to coimmunostaining analysis with antibodies specific to CEP250 (cyan) and acetylated tubulin (magenta) with and without daughter centriole association. **(E)** The number of cells with cilia was counted in *CEP250* KO cells with and without daughter centriole association. **(F)** The number of cells with centrosome/basal body CROCC signals was counted in *CEP250* KO cells with and without cilia in two different cell densities. **(G)** The *CEP250* KO cells were cultured in normal and serum-deprived media for 48 h

and subjected to coimmunostaining analysis with antibodies specific to CROCC (cyan) and acetylated tubulin (magenta). **(H)** The number of cells with centrosome/basal body CROCC signals was counted in *CEP250* KO cells cultured in two different cell densities. **(A, D, and G)** Scale bars, 10 µm; inset scale bars, 2 µm. **(B, C, E, F, and H)** More than 30 cells per group were counted in three independent experiments. Graph values are expressed as mean and SEM. Statistical significance was determined using one-way ANOVA with Tukey's post hoc test (*, $P < 0.05$; **, $P < 0.01$; ***, $P < 0.001$; n.s., not significant).

centrosome PCM1 levels were augmented in CEP72-depleted cells, irrespective of the *CEP250* and *CROCC* KO (Fig. 4, A–C). As previously reported, the cilia formation rate was reduced to some extent in the CEP72-depleted cells, which might be due to the improper delivery of key components for ciliary function (Fig. 4 D; Stowe et al., 2012). However, depletion of CEP72 in the *CEP250* and *CROCC* KO cells complemented the absence of intercentriolar/rootlet fibers with regard to the cilia formation rates as well as PCM1 accumulation in the centrosomes (Fig. 4,

B–D). These findings are consistent with the hypothesis in which intercentriolar/rootlet fibers facilitate cilia assembly through the accumulation of centriolar satellites near the centrosomes.

### The specific association of CROCC with PCM1 is essential for cilia assembly

We performed coimmunoprecipitation assays to determine the physical associations between the CROCC and centriolar satellite proteins. Since the endogenous CROCC protein is highly

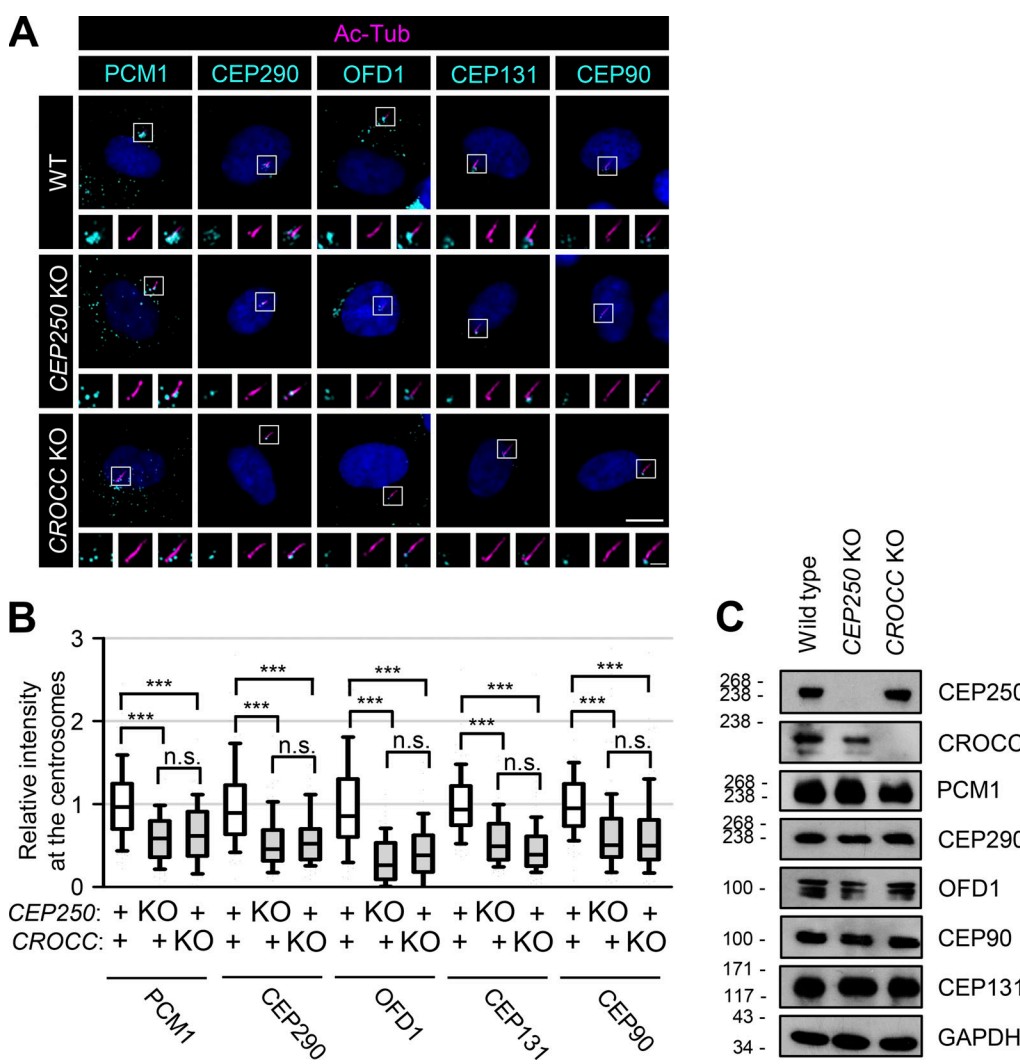

Figure 3. **Reduction of the centrosome/basal body levels of PCM1 in the *CEP250* and *CROCC* KO cells. (A)** The *CEP250* and *CROCC* KO RPE1 cells were cultured in a serum-deprived medium for 48 h to induce cilia assembly and coimmunostained with antibodies specific to acetylated tubulin (magenta), along with PCM1, CEP290, OFD1, CEP131, and CEP90 (cyan). Scale bar, 10 µm; inset scale bar, 2 µm. **(B)** Centrosome intensities of PCM1, CEP290, OFD1, CEP131, and CEP90 were determined. More than 30 cells per group were counted in three independent experiments. Within each box, the black center line represents the median value, the black box contains the interquartile range, and the black whiskers extend to the 10th and 90th percentiles. Statistical significance was determined using one-way ANOVA with Tukey's post hoc test (***, $P < 0.001$; n.s., not significant). **(C)** The *CEP250* and *CROCC* KO RPE1 cells were subjected to immunoblot analyses with antibodies specific to CEP250, CROCC, PCM1, CEP290, OFD1, CEP90, CEP131, and GAPDH. Source data are available for this figure: SourceData F3.

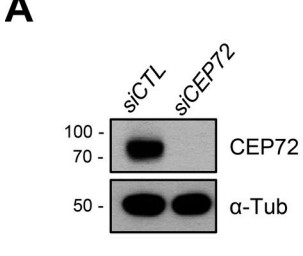

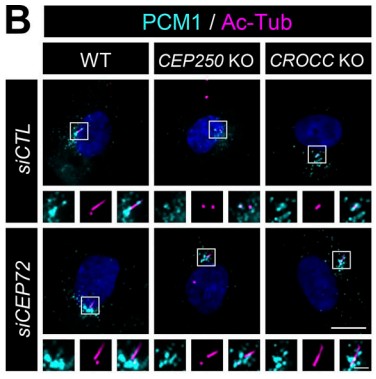

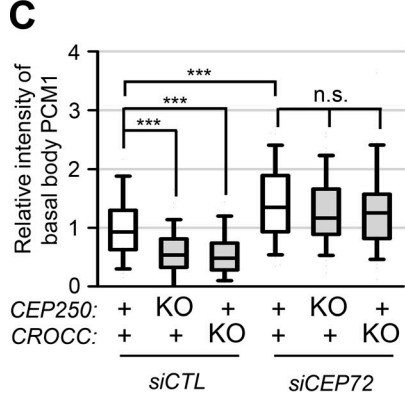

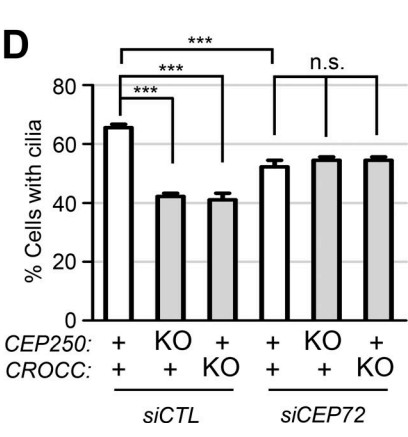

**Figure 4. Augmentation of the proportion of cells with cilia by CEP72 depletion. (A)** The CEP72-depleted RPE1 cells were cultured in a serum-deprived medium for 48 h and immunoblotted with antibodies specific to CEP72 and α-tubulin. **(B)** CEP72 was depleted in the CEP250 and CROCC KO cells and subjected to coimmunostaining analysis with antibodies specific to PCM1 (cyan) and acetylated tubulin (magenta). Scale bar, 10 µm; inset scale bar, 2 µm. **(C)** Intensities of PCM1 at the basal bodies were determined. Within each box, the black center line represents the median value, the black box contains the interquartile range, and the black whiskers extend to the 10th and 90th percentiles. **(D)** The number of cells with cilia was counted. Graph values are expressed as mean and SEM. **(C and D)** More than 30 cells per group were counted in three independent experiments. Statistical significance was determined using one-way ANOVA with Tukey's post hoc test (***, P < 0.001; n.s., not significant). Source data are available for this figure: SourceData F4.

insoluble, ectopic FLAG-CROCC was employed in this study (Yang et al., 2002). Both PCM1 and CEP131 were coimmunoprecipitated with FLAG-CROCC; however, CEP290 and OFD1 were not coimmunoprecipitated at all (Fig. 5 A). Reciprocal coimmunoprecipitation assays with the PCM1 antibody confirmed the physical interaction of PCM1 with FLAG-CROCC (Fig. 5 B). In CEP131-depleted cells, PCM1 was coimmunoprecipitated with FLAG-CROCC; however, in PCM1-depleted cells, CEP131 was not efficiently coimmunoprecipitated with FLAG-CROCC (Fig. 5 C). This suggests that a significant amount of CEP131 may indirectly be associated with FLAG-CROCC through PCM1.

To define a specific domain of CROCC that interacts with PCM1, we performed coimmunoprecipitation assays with the truncated mutants of FLAG-CROCC (Fig. S4). As summarized in Fig. 5 D, two independent sites at the N-terminal and C-terminal ends of CROCC were identified as binding sites for PCM1. We performed coimmunoprecipitation assays with FLAG-CROCC[303–1741] in which both the binding sites were truncated. As expected, endogenous PCM1 was coimmunoprecipitated with FLAG-CROCC[FL], but not with FLAG-CROCC[303–1741] (Fig. 5 E).

To investigate the functional significance of the CROCC-PCM1 interaction, we generated inducible stable lines of FLAG-CROCC and FLAG-CROCC[303–1741] in the CROCC KO RPE1 cells. Leaky expression of FLAG-CROCC appeared to be sufficient to rescue the CROCC knockout phenotype (Fig. S5, A and B). Super-resolution microscopic analysis revealed that intercentriolar fibers with FLAG-CROCC[303–1741] are indistinguishable from those with FLAG-CROCC[FL] (Fig. 6 A). However, PCM1 was not

concentrated at the basal bodies in the FLAG-CROCC[303–1741]-rescued cells (Fig. 6, B and C; and Fig. S5 C). The cilia formation rate was not restored in the FLAG-CROCC[303–1741]-rescued cells, either (Fig. 6, D and E). These results revealed that a physical association between CROCC and PCM1 is essential for primary cilia assembly as well as for centrosome accumulation of centriolar satellites.

Centriolar satellites are transported through microtubule networks (Dammermann and Merdes, 2002; Kubo and Tsukita, 2003). To determine whether microtubules are involved in the specific interaction of intercentriolar fibers with centriolar satellites, we treated nocodazole to the CROCC and CEP250 KO cells. We observed that the centrosome intensities of PCM1 in CROCC KO cells were reduced by nocodazole (Fig. 7, A and B). However, nocodazole did not affect the physical association of FLAG-CROCC with PCM1, suggesting that the physical interaction of centriolar satellites with intercentriolar fibers is independent of the microtubule network (Fig. 7 C).

We expressed the FLAG-CROCC proteins in CROCC KO cells and determined the subcellular distribution of the ectopic proteins. Accordingly, FLAG-CROCC was detected at the centrosomes while excess proteins were found to be located at the nuclear membrane (Fig. 7 D; Potter et al., 2017). Endogenous PCM1 followed the subcellular distribution of FLAG-CROCC at both the centrosome and the nuclear membrane (Fig. 7 D). Further, PCM1 at the nuclear membrane was not disturbed by nocodazole treatment (Fig. 7 D). Flag-CROCC[ΔR3], which lacks the third coiled-coil domain of CROCC, did not form intercentriolar fibers but was located at the nuclear membrane (Fig. 7 D; Ko et al., 2020). PCM1 also followed the cellular distribution of

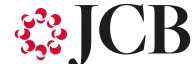

Figure 5. **Definition of PCM1-interacting regions in the CROCC protein. (A)** Lysates of the RPE1 cells expressing the ectopic FLAG-GFP and FLAG-CROCC proteins were immunoprecipitated with the FLAG antibody and subsequently immunoblotted with antibodies specific to FLAG, PCM1, CEP290, OFD1, and CEP131. The asterisk indicates non-specific band with the OFD1 antibody. **(B)** The same cell lysates were immunoprecipitated with the PCM1 antibody and subsequently immunoblotted with antibodies specific to PCM1, CROCC, and FLAG. Rabbit IgG was used as a negative control. **(C)** Endogenous PCM1 and CEP131 were depleted in stable RPE1 cells expressing FLAG-CROCC. The cell lysates were immunoprecipitated with the FLAG antibody and subsequently immunoblotted with antibodies specific to FLAG, PCM1, and CEP131. **(D)** Schematic of the truncated mutants of FLAG-CROCC. The interactions between the CROCC truncated mutants and endogenous PCM1 are summarized on the right. **(E)** Lysates of the stable cell lines expressing FLAG-CROCC$^{FL}$ and FLAG-CROCC$^{303–1741}$ were immunoprecipitated with the FLAG antibody and subsequently immunoblotted with antibodies specific to FLAG and PCM1. Source data are available for this figure: SourceData F5.

FLAG-CROCC$^{\Delta R3}$ in the cytoplasm (Fig. 7 D). Although nocodazole disturbed the cellular distribution of FLAG-CROCC$^{\Delta R3}$, it still colocalized with PCM1 (Fig. 7 D). FLAG-CROCC$^{303–1741}$ was detected at the centrosomes but not at the nuclear membrane; this is because it lacks a specific binding domain for nesprin1, which is located within the N-terminal domain of CROCC (Fig. 7 D; Potter et al., 2017). PCM1 was dispersed in the cytoplasm of FLAG-CROCC$^{303–1741}$-rescued cells (Fig. 7 D). Such findings strongly suggest that the physical association between CROCC and PCM1 is critical for the cellular distribution of centriolar satellites in cells.

## Definition of the CROCC-interacting region within the PCM1 protein

We generated *PCM1* knockout cell lines using the CRISR/Cas9 method and confirmed the absence of the PCM1 protein with immunoblot and immunostaining analyses (Fig. 8 A; and Fig. S6, A and B). As expected, the major centriolar satellite proteins, such as CEP290, OFD1, CEP131, and CEP90, were dispersed from the centrosomes, even if their expression levels were more or less unaffected in *PCM1* KO cells (Fig. S6, C–E; Wang et al., 2016). At the same time, the cilia formation rate was significantly reduced by more than fourfold in the *PCM1* KO cells (Fig. 8 B; Wang et al., 2016).

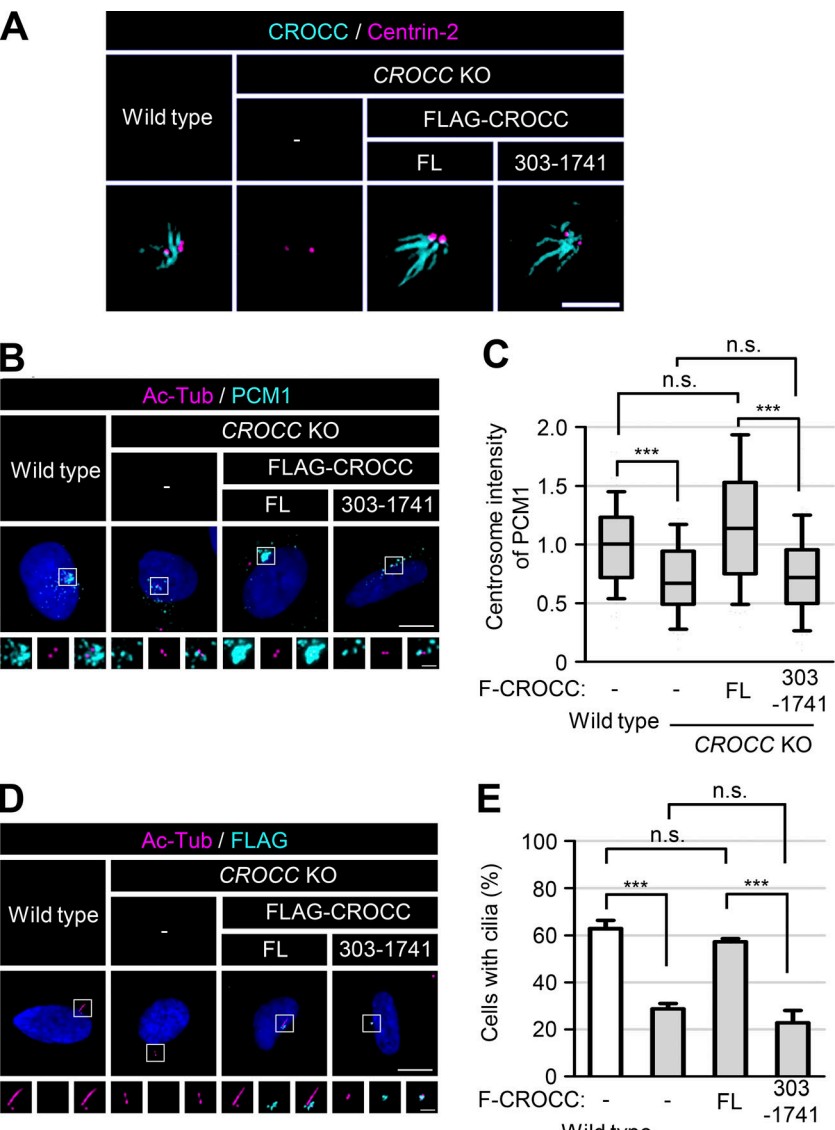

Figure 6. **Specific interaction of CROCC with PCM1 is essential for cilia assembly.** **(A)** FLAG-CROCC[FL] and FLAG-CROCC[303–1741] were stably expressed in the *CROCC* KO RPE1 cells. The cells were coimmunostained with antibodies specific to CROCC (cyan) and centrin-2 (magenta). The CROCC fibers were observed with a super-resolution microscope. Scale bar, 2 µm. **(B)** The cells were coimmunostained with antibodies specific to acetylaged tubulin (magenta) and PCM1 (cyan). **(C)** Intensities of PCM1 at the centrosomes were determined. Within each box, the black center line represents the median value, the black box contains the interquartile range, and the black whiskers extend to the 10th and 90th percentiles. **(D)** The cells were cultured in a serum-deprived medium for 48 h, and coimmunostained with antibodies specific to acetylated tubulin (magenta) and FLAG (cyan). **(E)** The number of cells with cilia was counted. Graph values are expressed as mean and SEM. **(B and D)** Scale bars, 10 µm; inset scale bars, 2 µm. **(C and E)** More than 30 cells per group were counted in three independent experiments. Statistical significance was determined using one-way ANOVA with Tukey's post hoc test (***, $P < 0.001$; n.s., not significant).

We generated the truncated mutants of FLAG-PCM1 and stably expressed them in *PCM1* KO cells (Fig. 8, C and D). All the truncated FLAG-PCM1 proteins were detected at the centrosomes except FLAG-PCM1[1201–2016] and FLAG-PCM1[Δ551–1200], suggesting that the 551–1,200 region of PCM1 is important for centrosome localization of PCM1 (Fig. 8 E). The centriolar satellite pattern of FLAG-PCM was clearly detected with the PCM1 antibody (Fig. S6 I). The cilia formation rates were completely rescued with the full-length FLAG-PCM1 and only partially with the truncated FLAG-PCM1 proteins that have the 551–1,200 region (Fig. 8 F). However, the FLAG-PCM1 proteins which lack the 551–1,200 region did not rescue the cilia formation rates at all (Fig. 8 F). This suggests that the 551–1,200 region of PCM1 is critical not only for centrosome accumulation but also for cilia assembly.

We performed coimmunoprecipitation assays to determine the physical association of PCM1 with CROCC. The truncated mutants of FLAG-PCM1 were coexpressed with GFP-CROCC in 293T cells and immunoprecipitated with the FLAG antibody. The results showed that the truncated FLAG-PCM1 proteins were expressed well and immunoprecipitated with the FLAG antibody (Fig. 9 A). Furthermore, GFP-CROCC was coimmunoprecipitated along with FLAG-PCM1[FL], FLAG-PCM1[551–1200], and FLAG-PCM1[551–2016] (Fig. 9 A). Since GFP-CROCC was not coimmunoprecipitated with FLAG-PCM1[Δ551–1200], it is likely that a CROCC-interacting domain resides within the 551–1,200 region of PCM1 (Fig. 9 A). We also performed reciprocal coimmunoprecipitation assays and observed that GFP-PCM1[FL] and GFP-PCM1[551–1200] were coimmunoprecipitated but GFP-PCM1[Δ551–1200] did not, supporting the conclusion that the 551–1,200 region of PCM1 contains a CROCC-interacting domain (Fig. 9 B).

### Colocalization of centriolar satellites at the intercentriolar/rootlet fibers in the kidney epithelial cells

Even if we observed the specific interactions between PCM1 and CROCC with the coimmunoprecipitation assays, we failed to

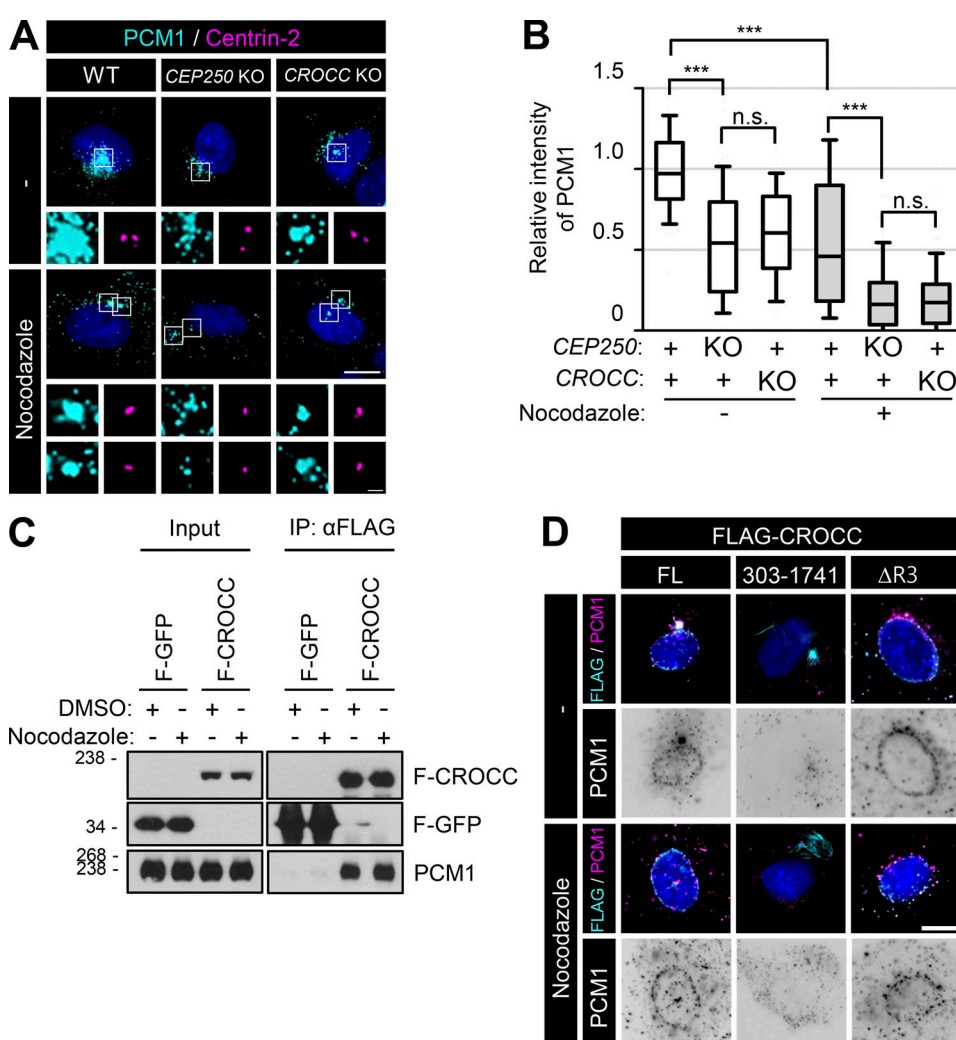

**Figure 7. Colocalization of PCM1 with subcellular CROCC. (A)** The *CEP250* and *CROCC* KO RPE1 cells were treated with 20 μM nocodazole for 2 h and subjected to coimmunostaining analysis with antibodies specific to PCM1 (cyan) and centrin-2 (magenta). **(B)** Intensities of PCM1 at the centrosome were determined. More than 30 cells per group were counted in three independent experiments. Within each box, the black center line represents the median value, the black box contains the interquartile range, and the black whiskers extend to the 10th and 90th percentiles. Statistical significance was determined using one-way ANOVA with Tukey's post hoc test (\*\*\*, P < 0.001; n.s., not significant). **(C)** The cells expressing the ectopic FLAG-CROCC protein were treated with 20 μM nocodazole for 2 h and subjected to immunoprecipitation analysis with the FLAG antibody, followed by immunoblot analyses with antibodies specific to FLAG and PCM1. **(D)** FLAG-CROCC$^{FL}$, FLAG-CROCC$^{303-1741}$, and FLAG-CROCC$^{\Delta R3}$ were stably expressed in the *CROCC* KO RPE1 cells. The cells were treated with 20 μM nocodazole for 2 h and coimmunostained with antibodies specific to FLAG (cyan) and PCM1 (magenta). **(A and D)** Scale bars, 10 μm; inset scale bar, 2 μm. Source data are available for this figure: SourceData F7.

observe specific localization of PCM1 at intercentriolar/rootlet fibers at least in the RPE1 cells. However, their interaction was hinted at in kidney tissues. We observed that the centriolar satellite proteins, such as CEP90 and CEP131, were coimmunostained with CROCC near the centrosomes as a rootlet pattern in most of the kidney tubular epithelial cells (Fig. 10, A and B). PCM1 was also colocalized at intercentriolar/rootlet fibers in half of the kidney epithelial cells (Fig. 10, A and B). However, we did not observe such colocalization of PCM1 and CEP90 at intercentriolar/rootlet fibers when the kidney cells were dissociated and cultured in vitro (Fig. 10, A and B). These results suggest that centriolar satellite proteins associate with the intercentriolar/rootlet fibers in cells at specific conditions, but their interactions may be loosened once the cells are forced to survive in culture.

## Discussion

In this work, we investigated the involvement of the intercentriolar/rootlet fibers in the cilia assembly process. We observed that CROCC, a main building block of the intercentriolar/rootlet fibers, specifically interacts with PCM1, a scaffold protein of centriolar satellites. Disruption of the CROCC–PCM1 interaction results in the dispersal of the centriolar satellites and reduction of the cilia formation rates as well. Based on the findings, we propose that the intercentriolar/rootlet fibers function as docking sites for centriolar satellites near the centrosomes/basal bodies and facilitate the cilia assembly process (Fig. 10 C).

Since the centrosome pairs are tightly associated with the intercellular microtubule network, they remain close to each other even in the absence of intercentriolar fibers in *CROCC* and

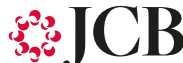

**Figure 8. Definition of the PCM1 domains for centrosome accumulation and cilia formation. (A)** The *PCM1* KO RPE1 cells were cultured in a serum-deprived medium for 48 h and coimmunostained with antibodies specific to PCM1 (cyan) and acetylated tubulin (magenta). **(B)** The number of cells with cilia was counted. **(C)** Schematic of the truncated mutants of FLAG-PCM1. Centrosome localizations of the PCM1 truncated mutants are summarized on the right. **(D)** Truncated mutants of FLAG-PCM1 were expressed in the *PCM1* KO RPE1 cells and subjected to immunoblot analyses with antibodies specific to FLAG and GAPDH. **(E)** The cells were coimmunostained with antibodies specific to FLAG (cyan) and γ-tubulin (magenta). **(F)** The number of cells with cilia was counted. **(A and E)** Scale bars, 10 μm; inset scale bars, 2 μm. **(B and F)** More than 30 cells per group were counted in three independent experiments. Graph values are expressed as mean and SEM. Statistical significance was determined using one-way ANOVA with Tukey's post hoc test (***, P < 0.001; n.s., not significant). Source data are available for this figure: SourceData F8.

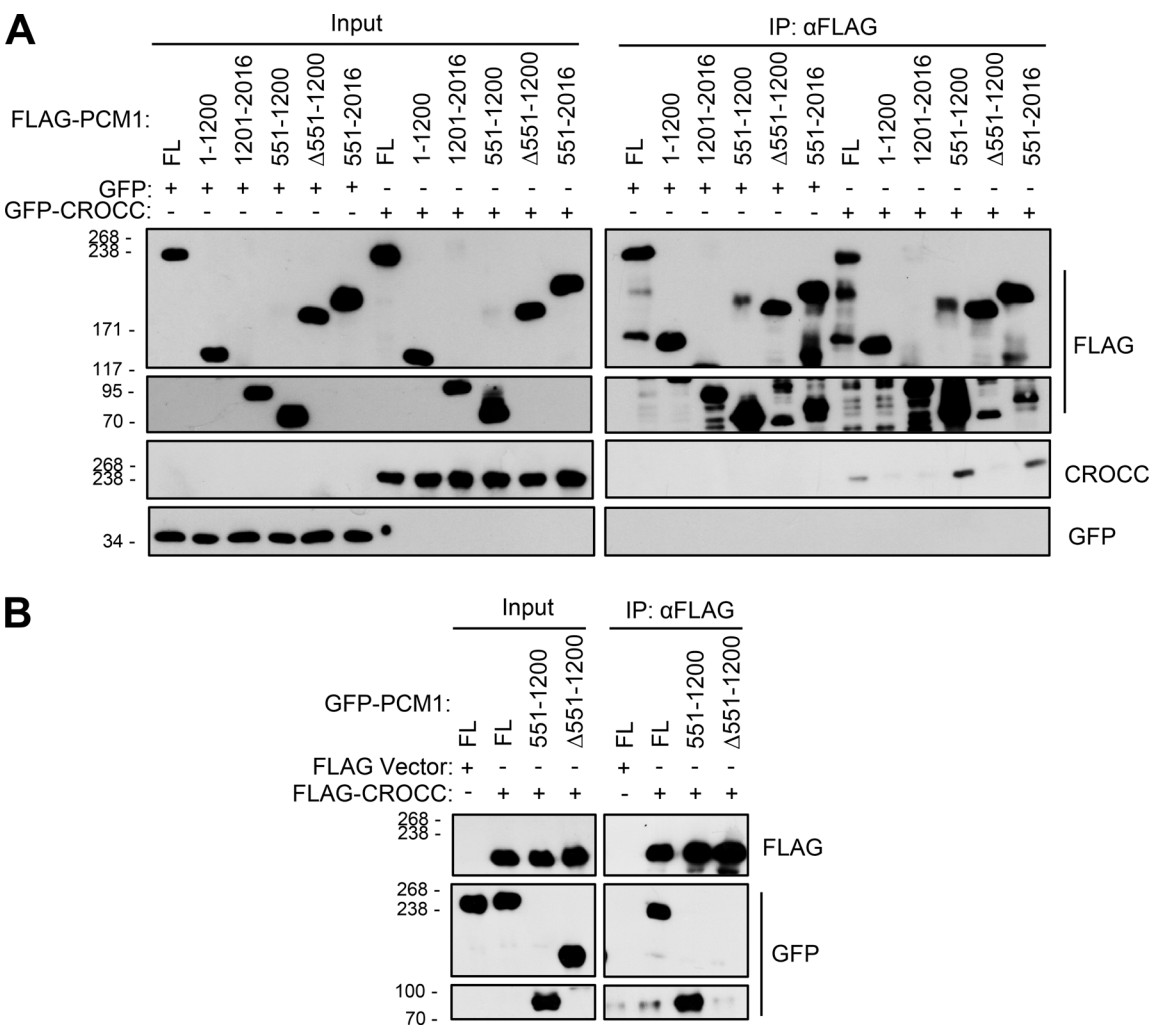

**Figure 9.** **Definition of the CROCC-interacting region in the PCM1 protein. (A)** Lysates of the 293T cells expressing the ectopic FLAG-PCM1 truncated proteins and GFP-CROCC were immunoprecipitated with the FLAG antibody and subsequently immunoblotted with antibodies specific to CROCC, GFP, and FLAG. **(B)** Lysates of the 293T cells expressing the ectopic FLAG-CROCC and GFP-PCM1 truncated proteins were immunoprecipitated with the FLAG antibody and subsequently immunoblotted with antibodies specific to FLAG and GFP. Source data are available for this figure: SourceData F9.

*CEP250* KO cells. However, we and others observed that centriolar satellites are dispersed from the centrosomes in the *CROCC* and *CEP250* KO cells (Fig. 3; Flanagan et al., 2017). In fact, the depletion of LRRC45, another component of the intercentriolar fibers, induces the dispersal of centriolar satellites and reduces cilia assembly (He et al., 2013; Kurtulmus et al., 2018). Depletion of CEP135, known to associate with CEP250 at the proximal end of the mother centriole, also makes centriolar satellites dispersed and the cilia formation rates reduced (Kim et al., 2008b; Hardy et al., 2014; Kurtulmus et al., 2016). Therefore, it is likely that the intercentriolar/rootlet fibers regulate the cellular dispersal of centriolar satellites.

Centriolar satellites are involved in cilia formation by regulating protein trafficking processes (Prosser and Pelletier, 2020; Aydin et al., 2020). Centriolar satellites travel through the microtubule networks to transport substances to areas near the centrosomes and cilia (Dammermann and Merdes, 2002; Kubo and Tsukita, 2003). If the microtubule network is disturbed by a destabilizing drug such as nocodazole, the centriolar satellites

are dispersed from the centrosomes, and cilia formation is consequently limited (Dammermann and Merdes, 2002; Kim et al., 2008a; Kim et al., 2012; Conkar et al., 2019). Other events, such as cellular stress or the absence of critical components, have been identified to trigger the dispersion of centriolar satellites (Villumsen et al., 2013; Gupta et al., 2015; Tollenaere et al., 2015). Once centriolar satellites are dispersed from the centrosomes, the cilia assembly is limited (Villumsen et al., 2013). However, it is also known from the *Pcm1* KO mouse study that centriolar satellites are dispensable for ciliogenesis in some cell types (Hall et al., 2023).

Cilia formation rates in the *PCM1* KO cells were reduced by one-fourth of the control cells (14 versus 62%; Fig. 8 B), which was more severe than those in the *CROCC* and *CEP250* KO cells (34 versus 61%; Fig. 2 B). This difference may be attributed to the role of PCM1 in delivering the ciliary cargoes to the centrosomes/basal bodies for ciliogenesis (Wang et al., 2016). Since a substantial amount of PCM1 signals are still detected at the centrosomes/basal bodies in *CROCC* and *CEP250* KO cells, a



Figure 10. **Localization of the centriolar satellite proteins at intercentriolar/rootlet fibers. (A)** Mouse kidney tissues and their primary culture cells were coimmunostained with the centriolar satellite protein (PCM1, CEP90, and CEP131; cyan) antibodies along with the CROCC (magenta) and acetylated tubulin (not shown) antibodies. Scale bar, 10 μm; inset scale bar, 2 μm. **(B)** The number of cells in which the centriolar satellite proteins were colocalized with CROCC at the intercentriolar/rootlet fibers was counted in the presence or absence of cilia. More than 200 cells per group were counted in two independent experiments. Graph values are expressed as mean and SEM. **(C)** Model. The centriolar satellites travel through the microtubule network to reach the intercentriolar/rootlet fibers near the centrosomes and cilia. The specific interaction between CROCC and PCM1 is essential for recruiting centriolar satellites near the centrosomes and cilia. The intercentriolar/rootlet fiber may serve as a docking site for centriolar satellites near the centrosomes/basal bodies. As a result, cargoes from the centriolar satellites are efficiently delivered to the vicinity of the centrosomes/basal bodies and facilitate the cilia assembly process.

fraction of cargoes may be delivered less efficiently to the centrosomes/basal bodies even in the absence of intercentriolar/rootlet fibers (Fig. 3 B). We also do not rule out the possibility that other centrosome proteins may contribute to the centrosome accumulation of the centriole satellites for the cilia assembly process.

Controversial results have been reported on cilia assembly in CROCC-deficient cells. Graser et al. (2007) observed no effects on cilia formation in the CROCC-depleted cell, although their knockdown efficiencies were not known. A significant reduction in cilia formation rates was reported in CEP250- and CROCC-depleted cells (Conroy et al., 2012; Nechipurenko et al., 2016). On the other hand, Turn et al. (2021) reported that the cilia formation rates increase in the *CROCC* KO mouse embryonic fibroblast cells. They argue that CROCC is inhibitory to ciliogenesis and, in the absence of CROCC, cilia formation is facilitated by the removal of CP110-CEP97 from the mother centrioles (Turn et al., 2021). In our experiments, we observed a reduction of the cilia formation rates in multiple lines of *CROCC* KO cells (Fig. S1). The difference may reside in cell type differences, as suggested by Turn et al. (2021). CROCC knockdown with two different siRNA transfection also reduced the cellular CROCC levels and revealed similar or even stronger phenotypes in PCM1 distribution and cilia formation rates as well as centriole disjunction (Fig. S7). However, we also noticed off-target effects, especially with *siRootletin#1*, which had previously been used in the CROCC knockdown experiments (Graser et al., 2007). For example, the number of cells with centriole disjunction increased further in *siRootletin#1*-treated cells in the *CROCC* KO cells, while the centrosome intensities of PCM1 further decreased (Fig. S7 C).

It was previously reported that *CEP250* KO does not affect cilia formation (Panic et al., 2015; Mazo et al., 2016; Flanagan et al., 2017). On the contrary, we observed a reduction of the cilia formation rates in the multiple clones of the *CEP250* KO cells (Fig. 2 B and Fig. S2). It is not clear why we observed a different outcome of *CEP250* KO from the others, but a part of the discrepancy may reside in the cell culture conditions. In a high cell density culture, the cilia formation rates of RPE1 cells increased, possibly due to mTORC1 activation and cell cycle exit (Takahashi et al., 2018). The cilia formation rate of the *CEP250* KO cells also increased nearly up to the control levels under the high-density culture condition (Fig. 2 F and Fig. S3 C). We also noticed that the centrosome localization of CROCC increased in cells cultured in high density (Fig. 2 F). If the RPE1 cells exited the cell cycle in a high cell density condition, CROCC might be placed at the centrosomes via interactions with unknown centrosome components. Our immunostaining images even hinted at a direct linkage of two centrioles with the CROCC fibers in the absence of CEP250 (Fig. 2 G and Fig. S2 G). In fact, CROCC has a similar domain structure to CEP250 (Yang and Li, 2006). *Drosophila rootletin* is the sole ortholog of the mammalian paralogs *CROCC/rootletin* and *CEP250/C-NAP1* (Chen et al., 2015; Styczynska-Soczka and Jarman, 2015). These results also support the model that intercentriolar/rootlet fibers are important for efficient cilia formation.

We revealed that centriolar satellites are dispersed in a cell when the specific interaction of PCM1 with CROCC was disrupted. Nonetheless, we failed to observe a distinct colocalization of the PCM1 proteins at the intercentriolar/rootlet fibers at least in the RPE1 cells. However, we happened to observe that signals of the centriolar satellite proteins and CROCC overlapped in kidney epithelial cells (Fig. 10, A and B). These results reveal that the centriolar satellite proteins may be bound to the intercentriolar/rootlet fibers in cells at specific tissues. Such strong associations were dramatically weakened when the same kidney cells were subjected to be cultured in vitro (Fig. 10, A and B). It is possible that the association of centriolar satellites with intercentriolar/rootlet fibers may be loosened due to certain cellular changes, such as cell division. It remains to be investigated how centriolar satellite accumulation at the intercentriolar/rootlet fibers is regulated in diverse cellular conditions.

Recently, an interaction between CROCC and PCM1 granules was suggested to occur in the multicilia assembly process (Zhao et al., 2021). The PCM1-containing granules, called fibrogranular materials, accumulate near the multiciliary area (Zhao et al., 2021). These materials presumably serve as cellular storage for centriolar and ciliary components for multicilia assembly (Zhao et al., 2021). Based on proximity labeling and immunostaining analyses, CROCC is linked to fibrogranular materials (Zhao et al., 2021). However, the exact roles of CROCC in fibrogranular materials during multicilia assembly should be investigated. Altogether, we believe that the CROCC–PCM1 interaction facilitates the unloading of ciliary materials near the cilia assembly area.

## Materials and methods
### Antibodies and plasmids
Rabbit anti-CROCC (HPA021191; IS, 1:200; IB, 1:300; Sigma-Aldrich), mouse anti-acetylated tubulin (T6793; IS, 1:200; Sigma-Aldrich), mouse anti-α-tubulin (T6199; IS, 1:1,000; IB, 1:10,000; Sigma-Aldrich), goat anti-FLAG (ab1257; IS, 1:500; IB, 1:2,000; Abcam), rabbit anti-CEP290 (ab84870; IS, 1:100; IB, 1:200; Abcam), rabbit anti-OFD1 (ab97861; IS, 1:100; IB, 1:100; Abcam), mouse anti-FLAG (F3165; IS, 1:2,000; IB, 1:20,000; Sigma-Aldrich), rabbit anti-γ-tubulin (ab11317; IS, 1:300; Abcam), rabbit anti-centrin-2 (04-1624; IS, 1:500; Millipore), mouse anti-γ-tubulin (ab11316; IS, 1:300; Abcam), rabbit anti-CEP68 (15147-1-AP; IS, 1:100; IB, 1:500; Proteintech), rabbit anti-CEP72 (A301-297A; IS, 1:500; IB, 1:500; Bethyl), and mouse anti-GAPDH (AM4300; IB, 1:10,000; Invitrogen) antibodies were purchased from commercial suppliers. Rabbit anti-PCM1 (Kim et al., 2012), rabbit anti-pericentrin (Kim and Rhee, 2011), rabbit anti-ninein (Lee and Rhee, 2015), rabbit anti-CEP250/C-NAP1 (Jeong et al., 2007), and rabbit anti-CEP90 (Kim and Rhee, 2011) polyclonal antibodies were prepared as described previously.

The human *CEP131* cDNA clone (Gene Bank accession number: AB029041) was purchased from the German Resource Center for Genome Research. We PCR-amplified the 341–1,008 fragment of the *CEP131* cDNA and subcloned it at the *Eco*RI site of the *pGEX-4T-1* vector (28-9545-49; Cytiva). The *pGST-CEP131$^{341–1008}$* plasmid was transformed into the *E. coli* BL21(DE3)pLysS strain. The bacteria were cultured to OD 0.8, treated with IPTG (isopropyl β-D-1-thiogalactopyranoside) to a final concentration of 0.5 mM for 4 h, and harvested. The GST-CEP131$^{341–1008}$ fusion protein

was purified using the GST beads (EBE-1041; Elpis Biotech). A pair of rabbits were immunized with a complete adjuvant (F5881-10ML; Sigma-Aldrich), which was combined with 150 µg of the GST-CEP131$^{341-1008}$ fusion protein and boosted with the same adjuvant containing the fusion protein in a 2-wk interval. 8 wk later, the rabbits were sacrificed, and the blood was drawn for collection of the CEP131 anti-sera. For the CEP131 antibody purification, 0.2 ml of the anti-serum was incubated for 2 h at room temperature with a PVDF (polyvinylidene fluoride) membrane on which 50 µg of the GST-CEP131$^{341-1008}$ fusion protein was blotted. The PVDF membrane was washed with TBST (Tris-buffered saline with 0.1% Tween 20) three times and incubated with 0.2 ml of an elution buffer (50 mM Tris pH 8.0, 100 mM NaCl, 4 mg/ml *L*-glutathione reduced) for 0.5 h. For neutralization, 20 µl of 1.5 M Tris (pH 8.8) was added to the CEP131 antibody eluent.

Alexa Fluor 488- and 594-conjugated secondary antibodies (Z25302; Invitrogen, Z25307; IS, 1:1,000; Invitrogen) were used for immunostaining. Anti-mouse IgG-HRP (A9044; IB, 1:1,000; Sigma-Aldrich), anti-rabbit IgG-HRP (AP132P; IB, 1:1,000; Millipore), and anti-goat IgG-HRP (SC-2056; IB, 1:500; Santa Cruz) were used as secondary antibodies for the immunoblot analyses. The CROCC/rootletin mutant subclones were previously described (Ko et al., 2020).

## Cell culture, transfection, and stable cell lines

The hTERT-RPE1 cells in our experiments were obtained from Dr. Kyung S. Lee (National Institutes of Health, Bethesda, MD, USA) (Soung et al., 2009). RPE1, HK2, IMCD3, and HEK293T cells were cultured in Dulbecco's modified eagle medium/Nutrient mixture F-12 (F12/DMEM) or DMEM supplemented with 10% FBS at 37°C under 5% $CO_2$. To induce cilia formation, the cells were transferred to a medium supplemented with 0.1% FBS and cultured for 48 h. We usually seed $2.5 \times 10^4$ cells per well in four-well dishes (1.96 cm²/well) to become $1.3 \times 10^4$ cells/cm². The RPE1 cells were transfected with siRNAs using RNAiMAX (Invitrogen) and with the plasmids using Lipofectamine3000 (Invitrogen) according to the manufacturer's instructions. The siRNAs used in this study were *siCTL* (5′-GCAAUCGAAGCUCGG CUACTT-3′), *siCROCC* (5′-AAGCCAGUCUAGACAAGGATT-3′), *siCEP250* (5′-CUGGAAGAGCGUCUAACUGAUTT-3′), *siPCM1* (5′-UCAGCUUCGUGAUUCUCAGTT-3′), *siCEP68* (5′-CACCCUCAA AUCACCUACUAATT-3′), *siCEP72* (5′-UUGCAGAUCGCUGGA CUUCAATT-3′), and *siCEP131* (5′-GCUAACAACAGGAGCAAC ATT-3′). To establish stable cell lines, CROCC, PCM1, and their mutants were subcloned into a *pcDNA5 FRT/TO* vector from Dr. Hyun S. Lee (Seoul National university, Seoul, Korea). For inducible expression, the RPE1 cells were transfected with the plasmids using Lipofectamine3000 (Invitrogen) and selected with G418 (5.09290; 400 µg/ml; Millipore) for 2 wk. For the immunoprecipitation assays, the plasmids were transiently transfected into HEK293T cells using the polyethyleneimine method.

## Generation of the knockout cell lines

gRNAs with high efficiency were designed using the CRISPR guide tool on the Benchling website (https://www.benchling.com/): *CROCC* gRNA1 (5′-AAACTGTCATGTGCTGGGTATGCAC-

3′ and 5′-CACCGTGCATACCCAGCACATGACA-3′) and gRNA2 (5′-CACCGATACTGTTTCATCCCCGGA-3′ and 5′-AAACTCCGG GGATGAAACAGTATC-3′) (Doench et al., 2016). *CEP250* gRNA1 (5′-CACCGAAGCTGAAGAACTCCCAGG-3′ and 5′-AAACCCTGG GAGTTCTTCAGCTTC-3′). *PCM1* gRNA1 (5′-CACCGAGCATTG GAAGTGATTCCCA-3′ and 5′-AAACTGGGAATCACTTCCAAT GCTC-3′). For CRISPR/Cas9 cloning, we used the plasmid *pSpCas9(BB)-2A-Puro (PX459) V2.0* (Plasmid #62988) as a gRNA vector backbone. The donor vector was digested with *Bbs*I and ligated with annealing gRNA using T4 DNA ligase (10481220001; Roche). RPE1 cells were transfected using Lipofectamine3000 (Invitrogen). After transfection, the cells were selected with 4 µg/ml puromycin (P8833; Sigma-Aldrich) for 48 h.

## Primary kidney cell culture

Primary kidney cells were isolated using a method adapted from Hakim et al. (2016). Briefly, freshly isolated kidneys from 8-wk-old mice were dissected and placed in ice-cold DMEM/F12. The capsule and ureters were removed, and the kidneys were minced using scalpel blades and incubated in 0.8 mg/ml collagenase type 2 (C6885; Sigma-Aldrich) and an equal concentration of soybean trypsin inhibitor (65035; Millipore) in HBSS at 37°C for 30 min with agitation. The tissue was allowed to settle, and the supernatant was collected in HBSS with 10% FBS. Collected fragmented tubules were centrifuged at 50 *g* for 7 min. The pellet was washed with DMEM/F12 and centrifuged at 50 *g* for 3 min. The pellet was resuspended in media and cells were grown on 12-mm coverslips coated with 100 µg/ml collagen (A1048301; Gibco). Cells were grown in a humidified incubator at 37°C in DMEM/F12 containing 0.1% FBS, 5 µg/ml transferrin (T2252; Sigma-Aldrich), 50 nM hydrocortisone (H0888; Sigma-Aldrich), and 5 µg/ml insulin (I6634; Sigma-Aldrich).

## Immunoprecipitation

The cells were lysed on ice for 15 min with lysis buffer (50 mM Tris-HCl pH 8.0, 5 mM EDTA, 150 mM NaCl, 0.5% Triton X-100, 1× protease inhibitor [P8340; Sigma-Aldrich], 0.5 mM phenylmethylsulfonyl fluoride, and 1 mM dithiothreitol). After centrifugation at 12,000 rpm for 15 min, the supernatants were incubated with FLAG-M2 Affinity Gel (A2220; Sigma-Aldrich) or Protein A Sepharose CL-4B (17-0780-01; Cytiva) for 90 min at 4°C. The beads were washed three times with lysis buffer and subjected to immunoblot analyses. All procedures were performed at 4°C.

## Immunoblot analyses

The cells were lysed on ice for 10 min with RIPA buffer (150 mM NaCl, 1% Triton X-100, 0.5% sodium deoxycolate, 0.1% SDS, 50 mM Tris-HCl at pH 8.0, 10 mM NaF, 1 mM $Na_3VO_4$, 1 mM EDTA, and 1 mM EGTA) containing a protease inhibitor cocktail (P8340; Sigma-Aldrich) and centrifuged with 12,000 rpm for 10 min at 4°C. The supernatants were mixed with 4×SDS sample buffer (250 mM Tris-HCl at pH 6.8, 8% SDS, 40% glycerol, and 0.04% bromophenol blue) and 10 mM DTT (0281-25G; Amresco). Mixtures were boiled for 5 min. The protein samples were loaded in SDS polyacrylamide gels (3% stacking gel and 4–10% separating gel), electrophoresed, and transferred to Protran

BA85 nitrocellulose membranes (10401196; GE Healthcare Life Sciences). The membranes were blocked with blocking solution (5% nonfat milk in 0.1% Tween 20 in TBS or 5% bovine serum albumin in 0.1% Tween 20 in TBS) for 2 h, incubated with primary antibodies diluted in blocking solution for 16 h at 4°C, washed four times with TBST (0.1% Tween 20 in TBS), incubated with secondary antibodies in blocking solution for 30 min, and washed again. To detect the signals of secondary antibodies, the ECL reagent (ABfrontier, LF-QC0101) and x-ray films (Agfa, CP-BU NEW) were used.

### Immunocytochemistry, immunohistochemistry, and image processing

The cells were cultured on 12-mm coverslips and fixed with cold methanol for 10 min or the PEM buffer (80 mM PIPES pH 6.9, 1 mM $MgCl_2$, 5 mM EGTA, and 0.5% Triton X-100). To detect primary cilia, microtubules were depolymerized via cold treatment for 60 min before fixation. The samples were blocked in PBST (PBS with 0.3% Triton X-100) with 3% BSA for 20 min, incubated with the primary antibodies for 1 h, and incubated with Alexa Fluor 488- and Alexa Fluor 594-conjugated secondary antibodies for 30 min (Life Technologies). 4′,6-Diamidino-2-phenylindole (DAPI) solution was used for DNA staining. The samples were mounted in ProLong Gold antifade reagent (P36930; Invitrogen) and observed using a fluorescence microscope (IX51; Olympus) equipped with a CCD (Qicam Fast 1394; Qimaging) camera using PVCAM (version 3.9.0; Teledyne Photometrics). We also used a super-resolution microscope (ELYRA PS.1; Carl Zeiss) for imaging CROCC at the centrosomes.

For immunohistochemistry, the kidney was fixed with 4% paraformaldehyde, embedded in paraffin blocks, and sectioned. The rehydrated sections were subjected to antigen retrieval in pH 9.0 Tris-EDTA buffer (10 mM Tris and 1 mM EDTA) in a microwave for 25 min, permeabilized in 0.1% PBST, blocked in 3% BSA in PBST, and incubated with the primary antibodies for 24 h followed by the secondary antibodies for 1 h (Life Technologies). After nuclear staining with DAPI, the samples were observed using a confocal laser-scanning microscope system (LSM700; Carl Zeiss).

The images were analyzed using ImagePro 5.0 software (Media Cybernetics, Inc.). Images were saved as Adobe Photoshop 2021 (version 22.4.2). For super-resolution images, the samples were observed using a super-resolution microscope (ELYRA PS.1; Carl Zeiss). SIM processing was performed with ZEN software 2012, black edition (Carl Zeiss), and the images were analyzed using ZEN lite software (Carl Zeiss).

### Measurements and statistical analysis

Imaging was performed with an Olympus IX51 microscope equipped with a CCD (Qicam Fast 1394; Qimaging) camera using PVCAM (version 3.9.0; Teledyne Photometrics). The fluorescence intensity and ciliary length were measured using ImageJ 1.53e software (National Institutes of Health). The fluorescence intensity was quantified by assessing the cumulative intensity within a circular region (20 µm²) centered between the centrioles. Statistical significance was determined using an unpaired two-tailed t-test and one-way analysis of variance (ANOVA) on Prism 6 (GraphPad software). Box and whisker plots display the median as a black center line, the interquartile range within the black box, and whiskers extending to the 10th and 90th percentiles. Bar graphs represent values as mean and SEM. In one-way ANOVA, groups sharing the same letter were not significantly different according to Tukey's post hoc test. *P < 0.05; P value of unpaired two-tailed t test.

### Online supplemental material

Fig. S1 shows the generation and characterization of *CROCC* KO RPE1 cells. Fig. S2 shows the generation and characterization of *CEP250* KO cells. Fig. S3 shows culture conditions for cilia formation of the RPE1 cells. Fig. S4 shows coimmunoprecipitation analysis of endogenous PCM1 with ectopically expressed FLAG-CROCC fragments. Fig. S5 shows ectopic expression of FLAG-CROCC proteins in *CROCC* KO RPE1 cells. Fig. S6 shows the generation and characterization of *PCM1* KO RPE1 cells. Fig. S7 shows the effects of *siCROCC* on CROCC expression in the *CROCC* KO RPE1 cells.

## Acknowledgments

The authors are grateful to Rhee's lab members who provided critical comments on the study.

This work was supported by the National Research Foundation of Korea grant funded by the Korean government (Ministry of Science and ICT) (no. NRF-2019R1A2C22002726). Open Access funding provided by Seoul National University.

Author contributions: S. Ryu, D. Ko, and K. Rhee designed the research. S. Ryu and D. Ko performed most of the experiments, analyzed, and interpreted data, while B. Shin performed CROCC localization at the kidney cells (Fig. 10). All the authors participated in writing the manuscript, approved the final version to be published, and agreed to be accountable for all aspects of this study.

Disclosures: The authors declare no competing interests exist.

Submitted: 12 May 2021

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

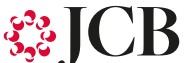

# Supplemental material

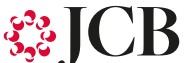

Figure S1. **Generation of the *CROCC* KO RPE1 cells. (A)** The *CROCC* KO RPE1 cells were coimmunostained with antibodies specific to γ-tubulin (magenta) and CROCC (cyan). **(B)** The *CROCC* KO cells were subjected to immunoblot analyses with antibodies specific to CROCC, PCM1, CEP250, CEP68, and GAPDH. **(C)** The number of cells with cilia was counted. **(D)** The distance between the centrioles in *CROCC* KO cells was determined after the treatment of 20 µM nocodazole for 2 h. **(E)** The *CROCC* KO cells were coimmunostained with antibodies specific to CEP68 (cyan) and centrin-2 (magenta). **(A and E)** Scale bars, 10 µm; inset scale bars, 2 µm. **(C and D)** More than 30 cells per group were counted in three independent experiments. Graph values are expressed as mean and SEM. Statistical significance was determined using one-way ANOVA with Tukey's post hoc test (*, P < 0.05; **, P < 0.01; n.s., not significant). Source data are available for this figure: SourceData FS1.

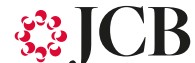

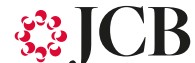

Figure S2. **Generation of the *CEP250* KO RPE1 cells. (A)** The *CEP250* KO RPE1 cells were coimmunostained with antibodies specific to centrin-2 (magenta) and CEP250 (cyan). **(B)** The *CEP250* KO cells were subjected to immunoblot analyses with antibodies specific to CEP250, PCM1, CROCC, CEP68, and GAPDH. **(C)** The number of cells with cilia was counted. **(D)** The number of cells with centriole disjunction (>2 μm) was counted. **(E)** The *CEP250* KO cells with and without cilia were coimmunostained with antibodies specific to CROCC (cyan) and acetylated tubulin (magenta). **(F)** The number of cells with centrosome/basal body CROCC signals was counted in *CEP250* KO cells with and without cilia. **(G)** The *CEP250* KO cells were cultured in serum-deprived medium for 48 h to induce cilia assembly and subjected to coimmunostaining analysis with antibodies specific to CROCC (cyan) and acetylated tubulin (magenta). **(H)** The number of cells with CROCC fibers was counted in cells. **(A, E, and G)** Scale bars, 10 μm; inset scale bars, 2 μm. **(C, D, F, and H)** More than 30 cells per group were counted in three independent experiments. Graph values are expressed as mean and SEM. Statistical significance was determined using one-way ANOVA with Tukey's post hoc test (**, P < 0.01; ***, P < 0.001; n.s., not significant). Source data are available for this figure: SourceData FS2.

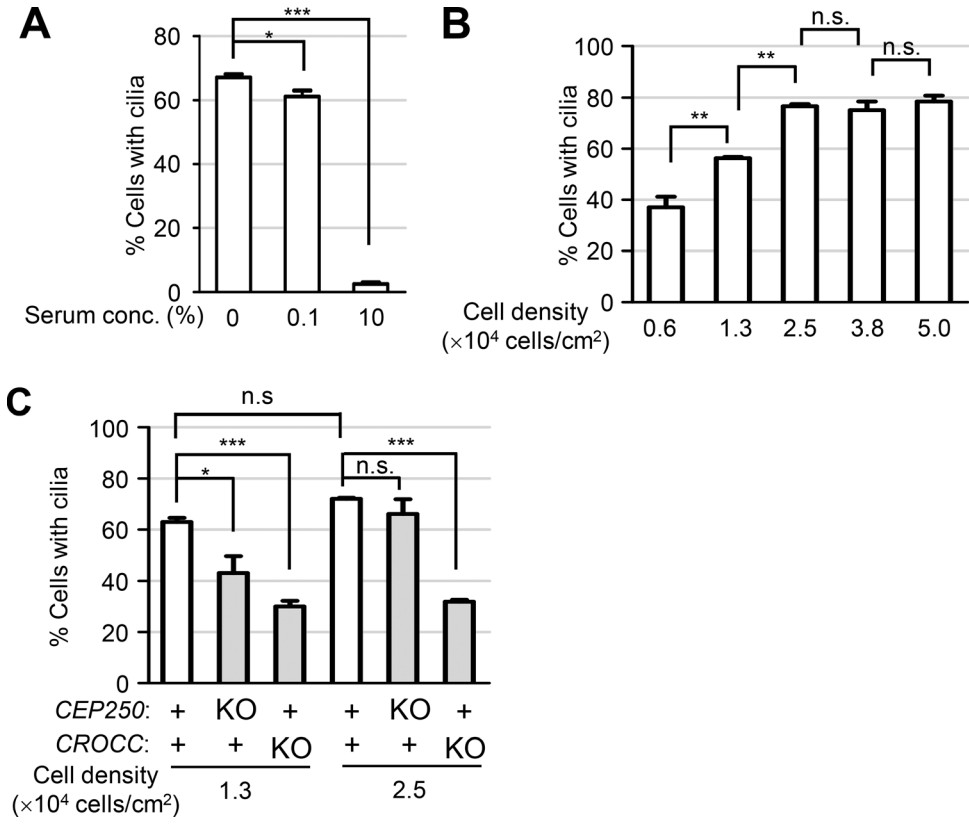

Figure S3. **RPE1 cell culture conditions for cilia formation. (A)** The number of cells with cilia was counted in RPE1 cells cultured in different serum concentrations. **(B)** The number of cells with cilia was counted in RPE1 cells cultured at different cell densities. **(C)** The number of cells with cilia was counted in CEP250 and CROCC KO cells cultured at different cell densities. **(A–C)** More than 30 cells per group were counted in three independent experiments. Graph values are expressed as mean and SEM. Statistical significance was determined using one-way ANOVA with Tukey's post hoc test (*, P < 0.05; **, P < 0.01; ***, P < 0.001; n.s., not significant).



Figure S4.   **Coimmunoprecipitation analysis of endogenous PCM1 with the FLAG-CROCC fragments. (A–D)** The FLAG-CROCC fragments were ectopically expressed in 293T cells and immunoprecipitated with the FLAG antibody followed by immunoblot analysis with the PCM1 and FLAG antibodies. Source data are available for this figure: SourceData FS4.



Figure S5. **Expression of FLAG-CROCC^FL and FLAG-CROCC^303–1741 in *CROCC* KO RPE1 cells. (A)** Ectopic expression of FLAG-CROCC^FL and FLAG-CROCC^303-1741 were induced with 1 µg/ml doxycycline for up to 4 h in the *CROCC* KO cells. The cells were coimmunostained with antibodies specific to FLAG (cyan) and Centrin-2 (magenta). Scale bar, 10 µm. **(B)** The cells were immunoblotted with antibodies specific to FLAG, CROCC, and GAPDH. **(C)** Intensities of PCM1 at the centrosomes were determined. More than 30 cells per group were counted in three independent experiments. Within each box, the black center line represents the median value, the black box contains the interquartile range, and the black whiskers extend to the 10th and 90th percentiles. Statistical significance was determined using one-way ANOVA with Tukey's post hoc test (**, $P < 0.01$; ***, $P < 0.001$; n.s., not significant). Source data are available for this figure: SourceData FS5.



**Figure S6.** **Generation and feature of *PCM1* KO RPE1 cells. (A)** The *PCM1* KO RPE1 cells were coimmunostained with antibodies specific to PCM1 (cyan) and centrin-2 (magenta). **(B)** The *PCM1* KO cells were subjected to immunoblot analyses with antibodies specific to PCM1 and GAPDH. **(C)** The *PCM1* KO cells were subjected to immunoblot analyses with antibodies specific to PCM1, CEP290, OFD1, CEP90, CEP131, and GAPDH. **(D)** The *PCM1* KO cells were cultured in a serum-deprived medium for 48 h and coimmunostained with antibodies specific to acetylated tubulin (magenta), along with PCM1, CEP290, OFD1, CEP131, and CEP90 (cyan). **(E)** Centrosome intensities of PCM1, CEP290, OFD1, CEP131, and CEP90 were determined. **(F)** The *PCM1* KO cells were coimmunostained with antibodies specific to centrin-2 (magenta), along with CEP250 and CROCC (cyan). **(G)** Centrosome intensities of CEP250 and CROCC were determined. **(H)** The *PCM1* KO cells were subjected to immunoblot analyses with antibodies specific to CEP250, CROCC, and GAPDH. **(I)** The FLAG-PCM1-expressing cells were coimmunostained with antibodies specific to PCM1 (cyan) and centrin-2 (magenta). **(A, D, F, and I)** Scale bars, 10 μm; inset scale bars, 2 μm. **(E and G)** More than 30 cells per group were counted in three independent experiments. Within each box, the black center line represents the median value, the black box contains the interquartile range, and the black whiskers extend to the 10th and 90th percentiles. Statistical significance was determined using one-way ANOVA with Tukey's post hoc test (***, P < 0.001; n.s., not significant). Source data are available for this figure: SourceData FS6.

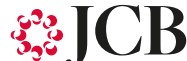

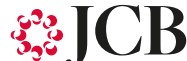

**Figure S7. Effects of *siCROCC* in the *CROCC* KO RPE1 cells. (A)** Intensities of CROCC at the centrosomes were determined after treatment of *siCROCC* in the *CROCC* KO cells. **(B)** The number of cells with centriole disjunction (>2 µm) was counted. **(C)** Intensities of PCM1 at the centrosomes were determined. **(D)** The number of cells with centriole disjunction (>2 µm) was counted after treatment of 20 µM nocodazole for 2 h. **(E)** Intensities of PCM1 at the centrosome were determined after treatment of 20 µM nocodazole for 2 h. **(A, C, and E)** Within each box, the black center line represents the median value, the black box contains the interquartile range, and the black whiskers extend to the 10th and 90th percentiles. **(B and D)** Graph values are expressed as mean and SEM. **(A–E)** More than 30 cells per group were counted in three independent experiments. Statistical significance was determined using one-way ANOVA with Tukey's post hoc test (*, P < 0.05; ***, P < 0.001; n.s., not significant).

