## [Peer Review File · The Journal of Cell Biology]

The inter-centriolar fibers function as docking sites of centriolar satellites for cilia assembly

Sungjin Ryu, Donghee Ko, Byungho Shin, and Kunsoo Rhee

Corresponding Author(s): Kunsoo Rhee, Seoul National University

Review Timeline:

Submission Date:	2021-05-12
Editorial Decision:	2021-07-06
Revision Received:	2023-08-09
Editorial Decision:	2023-08-29
Revision Received:	2023-12-07
Editorial Decision:	2023-12-11
Revision Received:	2023-12-15

Monitoring Editor: Karen Oegema

Scientific Editor: Dan Simon

Transaction Report:

DOI: <https://doi.org/10.1083/jcb.202105065>

July 6, 2021

Re: JCB manuscript #202105065

Prof. Kunsoo Rhee
Seoul National University
Department of Biological Sciences
1 Gwanak-ro, Gwanak-gu
Seoul 08826
Korea, Republic of (South Korea)

Dear Prof. Rhee,

Thank you for submitting your manuscript entitled "The inter-centriolar fibers are essential docking sites of centriolar satellites for cilia assembly" to Journal of Cell Biology. Unfortunately, after an assessment of the reviewer feedback, our editorial decision is against publication in JCB.

As you will see, your manuscript has now been read by two expert reviewers. Unfortunately, both reviewers were in agreement that although the prior literature on roles for rootletin in ciliogenesis is confusing, this study did not discuss or convincingly resolve these prior discrepancies. Instead the apparent differences between the knockout and siRNA knockdown results seemed to enhance the confusion. Overall, the reviewers felt that the advance made by the work with respect to rootletin and its connection to centriolar satellites and ciliogenesis is currently too limited to merit publication in the JCB. We hope that the reviewers' comments will be helpful in revising your manuscript for submission elsewhere.

Unfortunately I do not have the level of reviewer support that I would need to proceed further with the paper. I do realize that significant further work and expansion might convincingly address some of these issues, but I am hesitant to encourage you to work towards the aim of further consideration at JCB. The level of reviewer criticism makes it impossible for me to guarantee that we will be able to invite resubmission, even after revision. Therefore, it does seem that it will be best for you to consider another journal for this work. Our journal office will transfer your reviewer comments to another journal upon request.

I am sorry our decision is not more positive, but hope that you find the reviews constructive. Of course, this decision does not imply any lack of interest in your work and we look forward to future submissions from your lab.

Thank you for your interest in Journal of Cell Biology.

Sincerely,

Karen Oegema, PhD
Monitoring Editor
Journal of Cell Biology

Dan Simon, PhD
Scientific Editor
Journal of Cell Biology

Reviewer #1 (Comments to the Authors (Required)):

Ko, Ryu and Rhee here explore the roles of rootletin in ciliogenesis in RPE1 cells. Rootletin is a key component of the ciliary rootlet that also helps to maintain centriolar cohesion, but its functions in ciliogenesis are not clear.

The authors demonstrate an increase in rootlet signals at basal bodies compared to the signals at the base of centrosomes and then a decline in ciliation in rootletin-deficient cells after siRNA, along with a centriolar splitting phenotype consistent with previous publications on the topic. They show altered centriolar satellite recruitment in rootletin- and C-NAP1 depleted cells, as well as a reduction of rootletin in C-NAP1-depleted cells. Rescue of ciliogenesis and satellite density in rootletin-depleted cells is achieved by the co-depletion of CEP72. The authors define key regions of rootletin required for interaction with PCM1 and demonstrate that loss of N- and C-terminal interacting regions causes a loss of ciliogenesis capacity; furthermore, a centrosome-peripheral localisation of PCM1 depends on the rootletin interaction regions.

Key points

The data in the paper are clearly presented and well controlled; the experiments are appropriately performed. However, while the links to PCM1 are interesting and indicate a potential mechanism by which rootletin contributes to regulating cilia, there is relatively limited new information beyond this particular observation. Although the topic is of broad interest, it is not clear that the present submission provides either a 'definitive observation of outstanding interest' (as a Report) or a 'comprehensive analysis providing novel and significant mechanistic insight' (as an Article).

The literature on roles for rootletin in ciliogenesis is confusing: Graser et al (JCB 2007: PMID: 17954613) saw no strong impact of its depletion, while Conroy et al. (Cell Cycle 2012 PMID: 23070519) and Nechipurenko et al. (Dev Cell 2016 PMID: 27623382) both described declines in ciliogenesis in RPE1 cells upon rootletin depletion, similar to the data presented here. Turn et al. (MBoC 2021 PMID: 33596093) observed an opposing phenotype, of increased ciliary frequency in MEFs. These previous observations are not discussed in sufficient detail in this manuscript and it is unclear how the current submission is to be viewed alongside the literature. Data from *Drosophila* that explore cilia in rootletin-deficient neurons (Styczynska-Soczka and Jarman 2015 Cilia PMID: 26140210; Chen et al. 2015 JCB PMID: 26483560) should also be considered.

The limited impact of C-NAP1 depletion on ciliation (also described by Graser et al JCB 2007: PMID: 17954613; Panic et al. 2016 PLoS Genet PMID: 26001056; Flanagan et al. MBoC PMID: 28100636) does not fit well with the authors' model. Although they suggest that a residual rootlet structure may be sufficient to allow ciliogenesis, they do not test this idea, so that it remains a potentially important discrepancy.

Specific points

1. The legend to Figure 1 is potentially misleading- the rootletin/ CEP68 signals are not seen 'in cilia', so this should be rephrased. Similarly, a basal body is defined by its supporting a cilium, so this should be rephrased to distinguish between basal bodies (+ cilium) and centrosomes (- cilium).
2. The HGNC gene that encodes rootletin is CROCC and this should be used throughout. Similarly, CEP250 should be used for the gene that encodes C-NAP1.
3. Datapoints should be shown for Figs. 2D and 2H, or at least a box and whisker plot for these data.
4. It is unclear how linked/ split centrioles were determined/ defined in Fig. 2E.
5. The control in Fig. S1 is a crucial siRNA control, so should be included in Fig. 2.
6. The labelling within Fig 4 A, C, D appears to be incorrect as the double depletions are not labelled and the controls are mislabelled.
7. Does the CEP72 depletion rescue the rootletin KO phenotype?
8. Size markers should be provided for all immunoblot panels.
9. The conclusion that CEP131 does not interact with rootletin in PCM1-depleted cells is overstated; there is clearly some CEP131 detected. Even though this is equivalent to the amount seen in CEP131-depleted cells, it cannot be excluded that there is a limited interaction, so this conclusion should be toned down.
10. p6. The sentence, 'The coimmunoprecipitation assays performed with FLAGrootletin303-1741, in which both binding sites were deleted' should be rephrased.
11. The precise sourcing of the cells should be fully specified, particularly for the RPE1s and the FRT/TO derivative line.
12. As a minor point, for clarity, it would be helpful if the CRISPR-generated rootletin null (CROCC null) cells were referred to throughout as 'knockout', 'null' or 'KO', rather than as 'deleted'- this is very close to 'depleted' and risks confusing the reader.

Reviewer #2 (Comments to the Authors (Required)):

Review of The inter-centriolar fibers are essential docking sites of centriolar satellites for cilia assembly by Ko et al. Rootletin, a conserved centrosomal protein, is a component of the ciliary rootlets and is part of the G1 linker structure that connects the proximal ends of parental centrioles. In this manuscript Rhee and colleagues uncover an interaction between Rootletin and the centriolar satellite component PCM1, and conclude that Rootletin-dependent clustering of PCM1 near centrosomes is vital for cilia formation.

Although the molecular interaction between PCM1 and Rootletin is potentially important, I am not convinced that in its current form this manuscript is well-suited for JCB. The paper seems unfinished at places and the conclusions are not fully supported by the data. The manuscript provides new insights, but the overall advance it represents is relatively modest. In particular, contribution of PCM1 and satellites to ciliogenesis has been covered in several studies. Recently Rootletin was found to be included in PCM1 condensates in multiciliated cells. Furthermore, when c-NAP1, a component of the G1 linker, is deleted in RPE1 cells, cells display reduced density of centriolar satellites along with detachment of Rootletin from centrioles. Thus, the impact of Rootletin depletion on satellite distribution is not unexpected.

The manuscript is clearly written, and figures are well organised and presented. Additional proofreading would have been useful as I noted a number of mistakes in the legends and figure panels as described below.

Specific points:

1. The majority of the main figures depict data obtained with siRNA experiments (apart from Fig 6A-D), with results from Rootletin KO cells shown in the supplementary figures. I would have preferred seeing these data side-by-side, especially because a single siRNA is used throughout the experiments, which is inadequate, whereas two independent clones are analysed for the KO. An important difference I noted between si and KO of Rootletin is their impact on PCM1 distribution; it appears that relative intensity of PCM1 decreases only by ~40% in the KO compared to ~80% in siRootletin cells. In contrast, the effect on ciliogenesis seems to be very similar, both halving cilia formation (reducing from ~70% to ~35%). According to the literature, PCM1 deletion in RPE1 cells causes an almost complete loss of cilia, so it is strange that a 2-fold difference in PCM1 levels fail to differentially impact ciliogenesis. This discrepancy needs to be discussed and additional control experiments should be considered. For instance, the authors could test if siRootletin exacerbates PCM1 dispersion in KO cells, which would argue for an off-target effect.

2. Fig 3: It is not clear what the authors mean by 'concentrated' vs 'dispersed' PCM1 distribution. There are no images to explain these terms and I could not find any mention of how and where intensities were measured either in text, figure legends or methods. Fig 3A shows only the base of the cilium - it would be useful to include images of whole cells so that the reader can see satellite levels and morphology across the cell. It would be preferable to measure PCM1 intensities both in close proximity of centrosomes and a broader area surrounding the centrosome (similar to previous publications by the Pelletier lab) -- reporting these two values will provide a better insight into what is happening to PCM1 in these cells.

3. The title claims that "The inter-centriolar fibers are essential docking sites of centriolar satellites for cilia assembly ". I do not think the paper conclusively shows this, especially in light of the caveats listed in point 1. The authors remove over 500aa from Rootletin, and so one cannot exclude that other interactions of Rootletin are also lost and these contribute to the ciliation defect. Differences in PCM1 distribution between siRNA and KO may suggest a scenario where Rootletin contributes to cilia formation beyond tethering PCM1/satellites. Perhaps, the title could be toned down to reflect better the evidence provided in the paper.

4. Previous work showed that CEP72 depletion leads to an increase in centrosomal PCM1/satellites and reduction in their cytoplasmic pool. The data here indicates that this occurs independently of Rootletin, and once PCM1 accumulates at/around centrosomes, Rootletin is also dispensable for primary cilia formation. Can these results be confirmed by siCEP72 in the KO cells?

5. Fig. 4A-D: throughout the figure the pluses and minuses are mixed up under the graphs, it is a copy-paste error. Last of 4 samples should be co-depletion but instead there are two Cep72 sirnas and no control without siRootletin.

6. Fig S4B and S4C: on S4C 303-520 strongly interacts with PCM1 and on S4B it does not interact at all. Is the labelling correct?

7. Fig S3B: where is acetylated tubulin staining in the KO?

8. Figure 3 legend: typo: correct "and and for graphs in third line from bottom"

9. Figure 4 title: Augmentation of the cilia formation rate There is no data about the rate of ciliogenesis (i.e. no time course), so the title should be changed accordingly. Rate of ciliation is also used in the text, again, please change this.

10. Last paragraph in results section: typo: nonocodazole

11. I do not understand this argument in the discussion:

"Therefore, a significant number of rootletin fibers might remain near the centrosome area of the C-NAP1-deleted cells, possibly in association with other subcellular organelles, such as the endoplasmic reticulum (Panic et al., 2015). However, in the C-NAP1- and CEP68-depleted cells, the inter-centriolar/rootletin fibers may be sufficiently cleared from the centrosome area (Fig. 3 F). "

Why would depletion clear out these filaments more than deletion? The Morrison lab demonstrated that their published results concerning a negative impact of siRNA-mediated c-Nap1 depletion on ciliogenesis was an off target effect and could not be replicated in the knockout (Flanagan et al 2017).

Dear Prof. Rhee,

Thank you for submitting your manuscript entitled "The inter-centriolar fibers are essential docking sites of centriolar satellites for cilia assembly" to Journal of Cell Biology. Unfortunately, after an assessment of the reviewer feedback, our editorial decision is against publication in JCB.

As you will see, your manuscript has now been read by two expert reviewers. Unfortunately, both reviewers were in agreement that although the prior literature on roles for rootletin in ciliogenesis is confusing, this study did not discuss or convincingly resolve these prior discrepancies. Instead the apparent differences between the knockout and siRNA knockdown results seemed to enhance the confusion. Overall, the reviewers felt that the advance made by the work with respect to rootletin and its connection to centriolar satellites and ciliogenesis is currently too limited to merit publication in the JCB. We hope that the reviewers' comments will be helpful in revising your manuscript for submission elsewhere.

Unfortunately, I do not have the level of reviewer support that I would need to proceed further with the paper. I do realize that significant further work and expansion might convincingly address some of these issues, but I am hesitant to encourage you to work towards the aim of further consideration at JCB. The level of reviewer criticism makes it impossible for me to guarantee that we will be able to invite resubmission, even after revision. Therefore, it does seem that it will be best for you to consider another journal for this work. Our journal office will transfer your reviewer comments to another journal upon request.

I am sorry our decision is not more positive, but hope that you find the reviews constructive. Of course, this decision does not imply any lack of interest in your work and we look forward to future submissions from your lab.

Thank you for your interest in Journal of Cell Biology.

Sincerely,

Karen Oegema, PhD

Monitoring Editor

Dan Simon, PhD

Scientific Editor

Reviewer #1 (Comments to the Authors (Required)):

Ko, Ryu and Rhee here explore the roles of rootletin in ciliogenesis in RPE1 cells. Rootletin is a key component of the ciliary rootlet that also helps to maintain centriolar cohesion, but its functions in ciliogenesis are not clear.

The authors demonstrate an increase in rootlet signals at basal bodies compared to the signals at the base of centrosomes and then a decline in ciliation in rootletin-deficient cells after siRNA, along with a centriolar splitting phenotype consistent with previous publications on the topic. They show altered centriolar satellite recruitment in rootletin- and C-NAP1 depleted cells, as well as a reduction of rootletin in C-NAP1-depleted cells. Rescue of ciliogenesis and satellite density in rootletin-depleted cells is achieved by the co-depletion of CEP72. The authors define key regions of rootletin required for interaction with PCM1 and demonstrate that loss of N- and C-terminal interacting regions causes a loss of ciliogenesis capacity; furthermore, a centrosome-peripheral localisation of PCM1 depends on the rootletin interaction regions.

Key points

The data in the paper are clearly presented and well controlled; the experiments are appropriately performed. However, while the links to PCM1 are interesting and indicate a potential mechanism by which rootletin contributes to regulating cilia, there is relatively limited new information beyond this particular observation. Although the topic is of broad interest, it is not clear that the present submission provides either a 'definitive observation of outstanding interest' (as a Report) or a 'comprehensive analysis providing novel and significant mechanistic insight' (as an Article).

We are grateful of the critical comments. Structural support of the centrosome pair and cilia may be the prime role of the inter-centriolar/rootlet fibers. As pointed out by the reviewer, studies with *Drosophila* rootletin revealed its involvement in neuronal functions (Styczynska-Soczka and Jarman 2015; Chen et al. 2015). In this manuscript, we expanded the roles of inter-centriolar fibers to regulation of cilia assembly. Centriolar satellites are important for transport of key molecules for ciliogenesis. In our work, we revealed that CROCC/rootletin, a key component of the inter-centriolar/rootlet linker, specifically interacts with PCM1, a main scaffold of centriolar satellites. Based on a series of experiments, we propose that the inter-centriolar fibers function as a platform for centriolar satellites near the centrosome. We believe that our proposal is novel in elucidating mechanisms how ciliary components are retained near the centrosomes/basal bodies for cilia assembly. In the revised manuscript, we performed all experiments with the KO cell lines, instead of the knockdown cells. We also defined the specific CROCC-interacting regions within the PCM1 protein, and proved its importance in cilia formation. Furthermore, we observed that the selected centriolar satellite proteins are colocalized at the inter-centriolar/rootlet fibers in the kidney epithelial cells. We now believe that, in the revised manuscript, we carried out a comprehensive analysis for providing a novel and significant insight in biological functions of inter-centriolar/rootlet fibers in cilia assembly.

The literature on roles for rootletin in ciliogenesis is confusing: Graser et al (JCB 2007: PMID: 17954613) saw no strong impact of its depletion, while Conroy et al. (Cell Cycle 2012 PMID: 23070519) and Nechipurenko et al. (Dev Cell 2016 PMID: 27623382) both described declines in ciliogenesis in RPE1 cells upon rootletin depletion, similar to the data presented here. Turn et al. (MBoC 2021 PMID: 33596093) observed an opposing phenotype, of increased ciliary frequency in MEFs. These previous observations are not discussed in sufficient detail in this manuscript and it is unclear how the current submission is to be viewed alongside the literature. Data from *Drosophila* that explore cilia in rootletin-deficient neurons (Styczynska-Soczka and Jarman 2015 Cilia PMID: 26140210; Chen et al. 2015 JCB PMID: 26483560) should also be considered.

We are grateful of a comprehensive list of references for CROCC regulation on cilia assembly. We also notice that the literatures on the subject have been confusing and controversial. To rule out a possibility of the off-target effects of siRNA and the effects of residual amounts of the target proteins, we generated the KO lines of *CROCC*, *CEP250*, and *PCMI*, and repeated the whole experiments with them. In the revised manuscript, we revealed that the cilia formation rates were significantly reduced in the *CROCC* and *CEP250* KO cell lines (Figure 2A, B). It is interesting that the mother and daughter centrioles hardly split in the *CROCC* KO cells and only slightly in the *CEP250* KO cells (Fig. 2 C). It is because additional linkages may work to hold the centrioles together. One of the forces should be the microtubule network, since the nocodazole treatment greatly augmented split of the centriole pairs (Fig. 2 C). This point was described in the revised manuscript.

Graser et al (2007) reported no strong impact on ciliogenesis in the *CROCC*-depleted cells, even though they did not show efficiency of rootletin knockdown with immunoblot analysis. Furthermore, when we performed knockdown experiments with the same siRNA sequence they had used, we observed off-target effects (Fig. S6). Therefore, we rather reserve interpretation of the results in Graser et al (2007). Conroy et al (2012) and Nechipurenko et al (2016) reported significant reduction in number of cells with cilia after depletion of *CROCC*, which is consistent with our results. Turn et al. (2021) reported that the cells with cilia increase in the *CROCC* KO mouse fibroblast cells. They argue that *CROCC* has inhibitory effects on ciliogenesis since CP110-CEP97 is removed from the mother centriole to facilitate cilia elongation in the absence of *CROCC*. They also added that inhibitory effects of *CROCC* on cilia formation may be observed at limited cell lines (Turn et al., 2021). We do not rule out the possibility that *CROCC* functions are cell type-specific in part, as suggested by the authors (Turn et al., 2021). This point was described in the revised manuscript.

As suggested, we referred the papers regarding *Drosophila* *CROCC* in the introduction and discussion sections of the revised manuscript (Styczynska-Soczka and Jarman 2015 Cilia PMID: 26140210; Chen et al. 2015 JCB PMID: 26483560).

The limited impact of C-NAP1 depletion on ciliation (also described by Graser et al JCB 2007: PMID: 17954613; Panic et al. 2016 PloS Genet PMID: 26001056; Flanagan et al. MBoC PMID: 28100636) does not fit well with the authors' model. Although they suggest that a residual

rootlet structure may be sufficient to allow ciliogenesis, they do not test this idea, so that it remains a potentially important discrepancy.

In order to make sure importance of CEP250 in cilia assembly, we generated the *CEP250* KO cells. We detected none of the CEP250-specific signals in both immunoblot and immunostaining analyses with the *CEP250* KO lines (Fig. S2, A and B). The CEP250 signals were detected at all the centrioles in the *CROCC* KO cells, whereas the CROCC signals were detected at a fraction of the centrosomes in the *CEP250* KO cells (Fig. 2, F and G). In fact, CROCC signals were detected at the basal bodies of the *CEP250* KO cells with cilia, but the centrosome CROCC signals were absent in many of the *CEP250* KO cells without cilia (Fig. 2, F and G). These results support the notion that the inter-centriolar fibers are important for proper formation of cilia in the *CEP250* KO cells.

Previous works showed that cilia formation rates were hardly affected by the *CEP250* KO (Panic et al., 2015; Mazo et al., 2016; Flanagan et al., 2017). It is not clear why their results are different from ours. One possibility may be that residual CEP250 at the proximal end of centrioles still recruits inter-centriolar/rootlet fibers. In fact, we frequently observed a slender CROCC fibers in the *CEP250* knockdown cells (data not shown). Another possibility may be that CROCC is directly linked to the proximal ends of the centrioles and forms inter-centriolar fibers without CEP250. CROCC shares a strong structural homology with CEP250 and *Drosophila* genome includes *rootletin* as a sole orthologue of the mammalian paralogs *CROCC/rootletin* and *CEP250/C-NAP1* genes (Yang and Li, 2006; Chen et al., 2015; Styczynska-Soczka and Jarman, 2015). This point was clearly described in the discussion section of the revised manuscript.

Specific points

1. The legend to Figure 1 is potentially misleading- the rootletin/ CEP68 signals are not seen 'in cilia', so this should be rephrased. Similarly, a basal body is defined by its supporting a cilium, so this should be rephrased to distinguish between basal bodies (+ cilium) and centrosomes (- cilium).

We were careful distinguishing the basal bodies from centrosomes. We also used centrosomes/basal bodies when it was hard to distinguish them.

2. The HGNC gene that encodes rootletin is CROCC and this should be used throughout. Similarly, CEP250 should be used for the gene that encodes C-NAP1.

As suggested, we renamed *rootletin* and *C-NAP1* to *CROCC* and *CEP250*, respectively.

3. Datapoints should be shown for Figs. 2D and 2H, or at least a box and whisker plot for these data.

Fig. 2, D and H of the original manuscript are now placed at the Fig. S1 D.

4. It is unclear how linked/ split centrioles were determined/ defined in Fig. 2E.

Centriole split was determined when two centrioles are 2.0 μm apart.

5. The control in Fig. S1 is a crucial siRNA control, so should be included in Fig. 2.

In the revised manuscript, we decided to perform all experiments with the KO cell lines. We generated multiple lines of KO cells for *CROCC*, *CEP250* and *PCM1*, and confirmed absence of the gene products with immunoblot and immunostaining analyses (Figs. S1, S2 and S5).

6. The labelling within Fig 4 A, C, D appears to be incorrect as the double depletions are not labelled and the controls are mislabelled.

In the revised manuscript, we depleted *CEP72* in the *CEP250* or *CROCC* KO cell lines.

7. Does the *CEP72* depletion rescue the rootletin KO phenotype?

As suggested, we depleted *CEP72* in the *CROCC/rootletin* KO cells. The results showed that the cilia formation rates increased in the *CEP72*-depleted cells. *PCM1* was accumulated to the centrosome.

8. Size markers should be provided for all immunoblot panels.

As suggested, we provided size markers for the immunoblot data.

9. The conclusion that *CEP131* does not interact with rootletin in *PCM1*-depleted cells is overstated; there is clearly some *CEP131* detected. Even though this is equivalent to the amount seen in *CEP131*-depleted cells, it cannot be excluded that there is a limited interaction, so this conclusion should be toned down.

We toned down the sentence as follow: This suggests that a significant amount of *CEP131* may indirectly associates with FLAG-*CROCC* through *PCM1* (Fig. 5 C).

10. p6. The sentence, 'The coimmunoprecipitation assays performed with FLAGrootletin303-1741, in which both binding sites were deleted' should be rephrased.

We rephrased the sentence as follow: We performed coimmunoprecipitation assays with FLAG-*CROCC*³⁰³⁻¹⁷⁴¹, in which both the binding sites for *PCM1* were truncated.

11. The precise sourcing of the cells should be fully specified, particularly for the RPE1s and the FRT/TO derivative line.

The FRT/TO RPE1 cells were obtained from Ximbio (UK). The information is indicated in the Materials and Methods section.

12. As a minor point, for clarity, it would be helpful if the CRISPR-generated rootletin null (*CROCC* null) cells were referred to throughout as 'knockout', 'null' or 'KO', rather than as 'deleted'- this is very close to 'depleted' and risks confusing the reader.

We believe that knockout (KO) may be the best way to describe the CRISPR-generated null cells. Therefore, we used the term 'KO' throughout the manuscript.

Reviewer #2 (Comments to the Authors (Required)):

Rootletin, a conserved centrosomal protein, is a component of the ciliary rootlets and is part of the G1 linker structure that connects the proximal ends of parental centrioles. In this manuscript Rhee and colleagues uncover an interaction between Rootletin and the centriolar satellite component PCM1, and conclude that Rootletin-dependent clustering of PCM1 near centrosomes is vital for cilia formation.

Although the molecular interaction between PCM1 and Rootletin is potentially important, I am not convinced that in its current form this manuscript is well-suited for JCB. The paper seems unfinished at places and the conclusions are not fully supported by the data. The manuscript provides new insights, but the overall advance it represents is relatively modest. In particular, contribution of PCM1 and satellites to ciliogenesis has been covered in several studies. Recently Rootletin was found to be included in PCM1 condensates in multiciliated cells. Furthermore, when c-NAP1, a component of the G1 linker, is deleted in RPE1 cells, cells display reduced density of centriolar satellites along with detachment of Rootletin from centrioles. Thus, the impact of Rootletin depletion on satellite distribution is not unexpected.

We appreciate for your critical comments. It was previously assumed that centriolar satellites are important for delivering ciliary components near the centrosomes/basal bodies. However, it lacks mechanistic explanation how the ciliary components are delivered from centriolar satellites to the centrosomes/basal bodies. In this manuscript, we propose that the inter-centriolar/rootlet fibers may function as docking sites of centriolar satellites in vicinity of centrosomes/basal bodies. Our proposal is based on the key observations of the specific CROCC-PCM1 interaction. In the revised manuscript, we added experimental evidence by defining a CROCC-interaction domain in the PCM1 protein (Figs. 8 and 9). The same domain turned out to be critical for normal cilia formation (Fig. 8 E). Furthermore, we observed that the centriolar satellite proteins were colocalized at the inter-centriolar/rootlet fibers at specific tissue cells, such as the kidney epithelial cells (Fig. 10). We now believe that the present experimental evidence fully supports our conclusion.

The manuscript is clearly written, and figures are well organised and presented. Additional proofreading would have been useful as I noted a number of mistakes in the legends and figure panels as described below.

Specific points:

1. The majority of the main figures depict data obtained with siRNA experiments (apart from Fig 6A-D), with results from Rootletin KO cells shown in the supplementary figures. I would have preferred seeing these data side-by-side, especially because a single siRNA is used throughout the experiments, which is inadequate, whereas two independent clones are analyzed for the KO. An important difference I noted between si and KO of Rootletin is their impact on PCM1 distribution; it appears that relative intensity of PCM1 decreases only by ~40% in the

KO compared to ~80% in siRootletin cells. In contrast, the effect on ciliogenesis seems to be very similar, both halving cilia formation (reducing from ~70% to ~35%). According to the literature, PCM1 deletion in RPE1 cells causes an almost complete loss of cilia, so it is strange that a 2-fold difference in PCM1 levels fail to differentially impact ciliogenesis. This discrepancy needs to be discussed and additional control experiments should be considered. For instance, the authors could test if siRootletin exacerbates PCM1 dispersion in KO cells, which would argue for an off-target effect.

While revising the manuscript, we realized that a *siRootletin* that has been used in the original manuscript has off-target effects. That is, the *siRootletin#1* treatment further increased the centriole disjunction rates and reduced the centrosome PCM1 levels even in the *CROCC* KO cells (Fig. S6). Therefore, we repeated all the experiments with the *CROCC* and *CEO250* KO cells in the revised manuscript. The results showed that the cilia formation rates (Fig. 2 B) and the centrosomal PCM1 levels (Fig. 3, A and B) were reduced to similar rates in both the *CROCC* and *CEP250* KO cells. These results confirmed that inter-centriolar fibers are important for cilia formation as well as for the centrosome accumulation of PCM1.

We also performed a series of experiments with the *PCM1* KO cells to examine specific interaction of *CROCC* with PCM1 during the cilia assembly process. As the reviewer pointed out, the cilia formation rate was greatly reduced by four folds in the *PCM1* KO cells (Fig. 8 B), in comparison to less than two folds in the *CROCC* and *CEP250* KO cells (Fig. 2 B). We interpret that the inter-centriolar/rootlet fibers facilitate delivery of the ciliary components from centriolar satellites to the centrosomes/basal bodies. The ciliary components from centriolar satellites may be delivered to the centrosomes/basal bodies less efficiently in the absence of inter-centriolar/rootlet fibers. This point was described in the discussion section of the revised manuscript.

2. Fig 3: It is not clear what the authors mean by 'concentrated' vs 'dispersed' PCM1 distribution. There are no images to explain these terms and I could not find any mention of how and where intensities were measured either in text, figure legends or methods. Fig 3A shows only the base of the cilium - it would be useful to include images of whole cells so that the reader can see satellite levels and morphology across the cell. It would be preferable to measure PCM1 intensities both in close proximity of centrosomes and a broader area surrounding the centrosome (similar to previous publications by the Pelletier lab) -- reporting these two values will provide a better insight into what is happening to PCM1 in these cells.

We agree that the 'concentrated' and 'dispersed' are subjective terms in describing PCM1 distribution in a cell. In the revised manuscript, we determined the PCM1 intensities at the centrosome/basal body area, which is rather objective in describing cellular distribution of centriolar satellite proteins. We also measured PCM1 intensities in both the close proximity of centrosomes and compensated them with the PCM1 intensities at surrounding area. We presented the whole cell images for PCM1 distribution in the revised manuscript (Figs. 3 A and 7 D).

3. The title claims that "The inter-centriolar fibers are essential docking sites of centriolar satellites for cilia assembly ". I do not think the paper conclusively shows this, especially in light of the caveats listed in point 1. The authors remove over 500aa from Rootletin, and so one cannot exclude that other interactions of Rootletin are also lost and these contribute to the ciliation defect. Differences in PCM1 distribution between siRNA and KO may suggest a scenario where Rootletin contributes to cilia formation beyond tethering PCM1/satellites. Perhaps, the title could be toned down to reflect better the evidence provided in the paper.

In the knockout-rescue experiments, we showed that CROCC³⁰³⁻¹⁷⁴¹ is able to form inter-centriolar fibers to link two centrioles together (Fig. 6A), but fails to accumulate PCM1 into the centrosome area (Fig. 6, B and C). Importance of the CROCC-PCM1 interaction in cilia formation was also examined with another knockout-rescue experiments with PCM1^{Δ551-1200} which lacks the CROCC-interaction region, reinforcing the conclusion (Fig. 8, E and F). Nonetheless, we toned down the title, as follow: The inter-centriolar fibers function as docking sites of centriolar satellites for cilia assembly.

4. Previous work showed that CEP72 depletion leads to an increase in centrosomal PCM1/satellites and reduction in their cytoplasmic pool. The data here indicates that this occurs independently of Rootletin, and once PCM1 accumulates at/around centrosomes, Rootletin is also dispensable for primary cilia formation. Can these results be confirmed by siCEP72 in the KO cells?

As suggested, we repeated the same experiments with the *CROCC* and *CEP250* KO cells. The results showed that depletion of CEP72 still enhances PCM1 levels near centrosomes of the KO cells (Fig. 4 C). At the same time, cilia formation rates also increased after the depletion of CEP72, irrespective of the presence or absence of CROCC and CEP250 (Fig. 4 D). These results suggest that accumulation of PCM1 itself is important for cilia assembly. Furthermore, it is likely that the accumulation of centriolar satellites near the centrosomes/basal bodies may be one of biological functions of the inter-centriolar/rootlet fibers.

5. Fig. 4A-D: throughout the figure the pluses and minuses are mixed up under the graphs, it is a copy-paste error. Last of 4 samples should be co-depletion but instead there are two Cep72 sirnas and no control without siRootletin.

In the revised manuscript, we performed all the experiments in Fig. 4 with the *CROCC* or *CEP250* KO lines and labeled them correctly.

6. Fig S4B and S4C: on S4C 303-520 strongly interacts with PCM1 and on S4B it does not interact at all. Is the labelling correct?

We mislabeled 303-502 at the 157-302 lanes in Fig. 3 C of the revised manuscript. We are sorry for the mistake.

7. Fig S3B: where is acetylated tubulin staining in the KO?

We used enhanced images of the acetylated tubulin signals at the centrioles for Fig. 6 B of the revised manuscript (Fig. S3B of the original manuscript).

8. Figure 3 legend: typo: correct "and and for graphs in third line from bottom"

Corrected as suggested.

9. Figure 4 title: Augmentation of the cilia formation rate There is no data about the rate of ciliogenesis (i.e. no time course), so the title should be changed accordingly. Rate of ciliation is also used in the text, again, please change this.

Rewrote as 'Augmentation of the proportion of cells with cilia'

10. Last paragraph in results section: typo: nonocodazole

Corrected as suggested.

11. I do not understand this argument in the discussion:

"Therefore, a significant number of rootletin fibers might remain near the centrosome area of the C-NAP1-deleted cells, possibly in association with other subcellular organelles, such as the endoplasmic reticulum (Panic et al., 2015). However, in the C-NAP1- and CEP68-depleted cells, the inter-centriolar/rootletin fibers may be sufficiently cleared from the centrosome area (Fig. 3 F). "

Why would depletion clear out these filaments more than deletion? The Morrison lab demonstrated that their published results concerning a negative impact of siRNA-mediated c-Nap1 depletion on ciliogenesis was an off-target effect and could not be replicated in the knockout (Flanagan et al 2017).

In the revised manuscript, we determined the centrosome levels of CROCC in the *CEP250* KO cells under cilia formation conditions. The results showed that CROCC signals were detected at the basal bodies when cilia were formed in the *CEP250* KO cells (Fig. 2, F and G). However, the centrosome CROCC signals were absent in many of the *CEP250* KO cells without cilia, supporting the notion that the inter-centriolar/rootlet fibers are important for proper formation of cilia in the *CEP250* KO cells (Fig. 2, F and G). It is possible that two centrioles are directly linked by the inter-centriolar/rootlet fibers in the absence of CEP250. In fact, CEP250 has a similar domain structure with CROCC and *Drosophila rootletin (Root)* is the sole orthologue of the mammalian paralogs *CROCC/rootletin* and *CEP250/C-Nap1* (Yang and Li, 2006; Chen et al., 2015; Styczynska-Soczka and Jarman, 2015). Therefore, it is possible that CROCC directly associates with the proximal end of centrioles in the absence of CEP250. This results also support the model that inter-centriolar fibers are important for efficient cilia formation. This point is described in the discussion section of the revised manuscript.

August 29, 2023

Re: JCB manuscript #202105065R-A

Prof. Kunsoo Rhee
Seoul National University
Department of Biological Sciences
1 Gwanak-ro, Gwanak-gu
Seoul 08826
Korea, Republic of (South Korea)

Dear Prof. Rhee,

Thank you for submitting your revised manuscript entitled "The inter-centriolar fibers function as docking sites of centriolar satellites for cilia assembly." The manuscript has been seen by the original reviewers whose full comments are appended below. While the reviewers are overall positive about the work in terms of its suitability for JCB, some important issues remain.

In light of the positive comments of the reviewers, we invite you to submit a revision to address the reviewers remaining concerns. Many of the reviewers' points can be addressed with textual and figure modifications. However, a key focus of the revision, which will likely require additional experiments, should be to address points 1 and 2 of Reviewer #1, which request that you help set your manuscript in the literature by resolving the apparent discrepancies between your results and those of three prior studies that have also characterized the effect of C-NAP1 knockouts on ciliogenesis.

Our general policy is that papers are considered through only one revision cycle; however, given that the suggested changes are relatively minor we are open to one additional short round of revision. Please submit the final revision along with a cover letter that includes a point by point response to the remaining reviewer comments.

Thank you for this interesting contribution to Journal of Cell Biology. You can contact me or the scientific editor listed below at the journal office with any questions, cellbio@rockefeller.edu or call (212) 327-8588.

Sincerely,

Karen Oegema, PhD
Monitoring Editor
Journal of Cell Biology

Dan Simon, PhD
Scientific Editor
Journal of Cell Biology

Reviewer #1 (Comments to the Authors (Required)):

Ryu, Rhee and colleagues examine the functions of the centriolar linker in regulating primary ciliogenesis. This is a substantial revision of previously-submitted work in which CRISPR knockouts of the genes of interest, CEP250/C-NAP1 and CROCC (encoding rootletin), as well as PCM1, provide a more unified study.

The authors show increased CROCC/rootletin signal at basal bodies and reduced ciliation frequency in CEP250/C-NAP1 and CROCC knockout RPE1 cells. They also demonstrate a reduction of centrosome-proximal signals of centriolar satellite components in these knockouts. siRNA knockdown of CEP72 caused increased PCM1 localisation to centrosomes in wild-type and C-NAP1 and CROCC nulls, and abrogated the reduced ciliogenesis phenotype in the knockout cells. The authors show pulldown experiments with fragments of rootletin that indicate the requirement for specific regions of the protein for PCM1 interaction and demonstrate that the PCM1 interaction regions of rootletin are necessary for it to facilitate ciliogenesis. Figure 7 shows that PCM localisation to centrosome-proximal regions requires both microtubules and centriolar linker proteins. They also test the regions of PCM1 needed for centrosome localisation and rootletin interaction. Finally, they examine colocalisation in mouse kidney cells of centriolar satellite and linker proteins.

Together, the study provides potential new insight into the interplay between the centriole linker structures and the centriolar satellites, specifically in the regulation of ciliogenesis. These findings could be of broad general interest and may open new avenues for the exploration of centrosome linker and centriole satellite functions. While the experiments are generally well

performed and controlled, there are several points that should be addressed:

1. A significant issue lies in positioning this study within the literature. 3 previous independent studies from different groups have reported knockouts of C-NAP1 in hTERT-RPE1 cells which no ciliation defect was seen (Panic et al., Mazo et al. and Flanagan et al., all cited here). The current MS. does not resolve this apparent discrepancy, which is of concern, and which may limit the assimilation of these findings by the field. At the very least, there should be a more detailed discussion to indicate why these findings are so different.

Points that could be considered are as follow:

- a. The percentage ciliogenesis described by Ryu et al. is lower than seen by the other groups (c. 60% as against 70-80% after 48h serum starvation). Are these hTERT-RPE1 cells or a different source/ clone? Where were they sourced from? This information should be in the Materials and Methods, as previously mentioned.
- b. The disruption of the CEP250 locus here is further 5' than in previous studies, so it is possible that a smaller C-NAP1 protein is generated (and thus a more pronounced phenotype observed). This might be tested directly in a relatively straightforward way- the authors could express (in their C-NAP knockouts) the [possible, predicted] proteins expressed in the other studies and test if these proteins allow ciliogenesis. One would not insist on this experiment, but a clear-cut result would be a very useful inclusion in the paper, if this were the case.
- c. It might also be feasible to test whether the cells from these other groups show a ciliogenesis phenotype under the conditions used in the Rhee lab. The precise conditions used for serum starvation (0% serum, 0.1% serum, etc.) should be specified in the Materials and Methods.
- d. Alternative splicing etc. might be an another idea that affects the outcomes; the extent to which the antibodies used in the various studies eliminate the possibility of residual expression of C-NAP1 might also be discussed.

2. While Ryu et al. suggest in the Discussion that residual CROCC may affect the phenotype in their cells, Figure S2E seems to show a less marked loss of CROCC protein from the C-NAP1-deficient centrioles than was described by Panic et al. and by Flanagan et al. However, this experiment may have been done after serum starvation, so may not be directly comparable. The treatment (starvation or not) should be clarified for each experiment. The question of how reduced the CROCC levels are in the currently-described C-NAP1 knockout should also be resolved, as this appears to be a significant difference from the previously-described effects of C-NAP1 loss on the centriolar linker.

3. Fig 1C appears to show an increase in CROCC over the serum starvation period, contrary to the authors' conclusion. A quantitation should be provided to clarify this issue.

4. The experiment shown in Fig S2D should be repeated and the relevant statistical information included.

5. The authors should describe the exonic position of the CRISPR cleavage sites used in the generation of the knockouts and the protein-level consequence of the disruptions. This is important to allow comparison with other work.

6. A uniform approach to labelling the statistical analyses should be taken throughout the paper (e.g., Fig. 3 uses * and 'n.s.', while Fig. 2 uses letters to indicate significance or otherwise). I suggest that the * and 'n.s.' is more conventionally used.

7. The cellular volume examined to define the centrosome-proximal intensities of the satellite proteins should be specified.

8. A rescue control should be shown for the CEP72 siRNA ciliation experiment in Figure 4.

9. The experiment shown in Fig. S3E is not explained clearly in the text and could be omitted as currently presented.

10. Details of the microscopy should be expanded to say whether images presented are from single sections or, if deconvolved images, how many sections were imaged; objective lens details should also be provided.

11. Additional work that addresses the roles of centriolar satellites (PCM1) in ciliogenesis should be considered, particularly in respect of (potential) cell type-specific aspects of the model: Hall et al. 2023 (PMID 36790165); Aydin et al. 2020 (PMID 32555591).

Reviewer #2 (Comments to the Authors (Required)):

The authors have addressed the majority of my comments and substantially improved the manuscript. I have a few minor points to make at this stage:

1) Fig 2G is not intuitive, especially the split boxes in the graph that show % of CROCC+ and CROCC- centrosomes in cells without cilia. According to the methods, groups with same letter are not significantly different but I don't see the point of comparing percentage of cells without cilia in CEP250 KO to percentage of cells with cilia in WT.

- 2) The degree of OFD1 reduction in basal bodies of CEP250KO and CROCC KO is surprising. While OFD1 associates with centriolar satellite, it has a large pool at distal appendages, where it is required for cilia formation. Given that CEP90 is still detectable, distal appendages probably still form in these cells, but the authors should highlight these points in the text.
- 3) In the discussion, it is stated "it is possible that CROCC directly associates with the proximal end of centrioles in the absence of CEP250". Being a rather important point, it would be straightforward to test this by high/super resolution microscopy.
- 4) The authors state "All the truncated FLAG-PCM1 proteins were detected at the centrosomes except FLAG-PCM1 1201-2016 and FLAG-PCM1 Δ 551-1200, suggesting that the 551-1200 region of PCM1 is important for centrosome localization of PCM1" I agree but why is there no satellite staining of FL PCM1 in PCM1 KO cells? Can any of the truncated versions form satellites?
- 5) The authors should highlight the caveat that the protein domains implicated in CROCC and PCM1 interaction are expansive and one cannot exclude other interactors contributing to the results.

Reviewer #1:

Ryu, Rhee and colleagues examine the functions of the centriolar linker in regulating primary ciliogenesis. This is a substantial revision of previously-submitted work in which CRISPR knockouts of the genes of interest, CEP250/C-NAP1 and CROCC (encoding rootletin), as well as PCM1, provide a more unified study.

The authors show increased CROCC/rootletin signal at basal bodies and reduced ciliation frequency in CEP250/C-NAP1 and CROCC knockout RPE1 cells. They also demonstrate a reduction of centrosome-proximal signals of centriolar satellite components in these knockouts. siRNA knockdown of CEP72 caused increased PCM1 localisation to centrosomes in wild-type and C-NAP1 and CROCC nulls, and abrogated the reduced ciliogenesis phenotype in the knockout cells. The authors show pulldown experiments with fragments of rootletin that indicate the requirement for specific regions of the protein for PCM1 interaction and demonstrate that the PCM1 interaction regions of rootletin are necessary for it to facilitate ciliogenesis. Figure 7 shows that PCM localisation to centrosome-proximal regions requires both microtubules and centriolar linker proteins. They also test the regions of PCM1 needed for centrosome localisation and rootletin interaction. Finally, they examine colocalisation in mouse kidney cells of centriolar satellite and linker proteins.

Together, the study provides potential new insight into the interplay between the centriole linker structures and the centriolar satellites, specifically in the regulation of ciliogenesis. These findings could be of broad general interest and may open new avenues for the exploration of centrosome linker and centriole satellite functions. While the experiments are generally well performed and controlled, there are several points that should be addressed:

1. A significant issue lies in positioning this study within the literature. 3 previous independent studies from different groups have reported knockouts of C-NAP1 in hTERT-RPE1 cells which no ciliation defect was seen (Panic et al., Mazo et al. and Flanagan et al., all cited here). The current MS. does not resolve this apparent discrepancy, which is of concern, and which may limit the assimilation of these findings by the field. At the very least, there should be a more detailed discussion to indicate why these findings are so different.

Points that could be considered are as follow:

The percentage ciliogenesis described by Ryu et al. is lower than seen by the other groups (c. 60% as against 70-80% after 48h serum starvation). Are these hTERT-RPE1 cells or a different source/ clone? Where were they sourced from? This information should be in the Materials and Methods, as previously mentioned.

We are grateful of your constructive critics on the manuscript. The hTERT-RPE1 cells in our experiments were obtained from Dr. Kyung S. Lee (National Institutes of Health, Bethesda, USA) (Soung et al., Dev Cell 16:539, 2009). We usually seed 2.5×10^4 cells per well in 4-well dishes (1.96 cm²/well). The cilia formation rates of our RPE1 cells have been always near 60%

in all papers published from my laboratory, beginning from Kim et al. (Plos One, 7:e48196, 2012).

As suggested, we determined cilia formation rates in diverse culture conditions, and found that the cell density is the most critical factor for cilia formation rates. We observed that about 60% of RPE1 cells formed cilia in 1.3×10^4 cells/cm², whereas nearly 80% of them formed cilia in 2.5×10^4 cells/cm² (Fig. S3 B). We were surprised that the cilia formation rates were affected by cell density, but learned that it had been known a long time ago. Takahashi et al. (JCS 131, jcs208769, 2018) recently reported that the proportion of ciliated cells increased, depending upon a rise in the density of RPE1 cells, probably due to the glucose deprivation and eventually to mTORC1 inactivation. We accept the points of Takahashi et al. (2018). When RPE1 cells were cultured in a high density, they would exit the cell cycle with little proliferation activity. As results, cilia may be formed in an extended number of the cells.

As suggested, we described this information in the Results and Discussion sections of the revised manuscript.

- b. The disruption of the CEP250 locus here is further 5' than in previous studies, so it is possible that a smaller C-NAP1 protein is generated (and thus a more pronounced phenotype observed). This might be tested directly in a relatively straightforward way- the authors could express (in their C-NAP knockouts) the [possible, predicted] proteins expressed in the other studies and test if these proteins allow ciliogenesis. One would not insist on this experiment, but a clear-cut result would be a very useful inclusion in the paper, if this were the case.

Panic et al. (2015) and Flanagan et al. (2017) targeted exon 15 and exon 8 of the *CEP250* gene, respectively, possibly resulting in truncated CEP250 mutants with 1-763 and 1-350 residues. Since we targeted exon 1 for generation of *CEP250* KO in RPE1 cells, we do not expect a C-terminal truncated CEP250 protein but maybe a long N-terminal truncated CEP250 mutant protein. As suggested, we stably rescued the *CEP250* KO cells with the full-length as well as truncated FLAG-CEP250 (1-350, 1-763, 351-2441, 764-2441) and determined the subcellular distributions and cilia formation rates (Attached Figure A1).

First, we determined the subcellular localization of the truncated mutants of FLAG-CEP250. FLAG-CEP250¹⁻³⁵⁰ and FLAG-CEP250¹⁻⁷⁶³ were not found at the centrosomes, as reported by Panic et al. (2015). On the other hand, FLAG-CEP250³⁵¹⁻²⁴⁴¹ and FLAG-CEP250⁷⁶⁴⁻²⁴⁴¹ as well as FLAG-CEP250^{FL} were detected at the centrosomes, suggesting that the C-terminal end of CEP250 is essential for centriole localization (Attached Figure A1B). Next, we determined the cilia formation rates in the *CEP250* KO cells rescued with the truncated mutants. The results showed that FLAG-CEP250³⁵¹⁻²⁴⁴¹ and FLAG-CEP250⁷⁶⁴⁻²⁴⁴¹ rescued the cilia formation rates to a normal range, but FLAG-CEP250¹⁻³⁵⁰ and FLAG-CEP250¹⁻⁷⁶³ did not (Attached Figure A1C). However, we also observed that FLAG-CEP250^{FL} did not rescued the cilia formation rates to a normal range, either (Attached Figure A1C). We should perform additional experiments to elucidate why the cilia formation rate of FLAG-CEP250^{FL} was lower than expected. Nonetheless, the results reveal that truncated mutants of CEP250 may rescue cilia formation rates as far as they are located at the centrosomes.

Attached Figure A1. Definition of the CEP250 domains important for cilia formation (A) Truncated mutants of FLAG-CEP250 were expressed in the *CEP250* KO RPE1 cells and subjected to immunoblot analyses with antibodies specific to FLAG and GAPDH. **(B)** The cells were coimmunostained with antibodies specific to FLAG (green) and γ -tubulin (red). Scale bar, 10 μ m. **(C)** The number of cells with cilia was counted in the cells rescued with the truncated mutants of FLAG-CEP250. More than 30 cells per group were counted in three independent experiments. Statistical

significance was determined using one-way ANOVA with Tukey's post hoc test (*, $P < 0.05$; n.s., not significant).

c. It might also be feasible to test whether the cells from these other groups show a ciliogenesis phenotype under the conditions used in the Rhee lab. The precise conditions used for serum starvation (0% serum, 0.1% serum, etc.) should be specified in the Materials and Methods.

As suggested, we determined cilia formation rates in RPE1 cells cultured in 0, 0.1 and 10% FBS. The results showed that the cilia assembled equally well in cells cultured in media with both 0 and 0.1% FBS, but not with 10% FBS (Fig. S3 A), suggesting that FBS may not be a key factor as far as its concentration is low. Rather, as we previously mentioned, the cell density is the most essential factor for cilia formation rates (Fig. S3, B and C). The cilia formation conditions are described in the Materials and Methods section in detail.

d. Alternative splicing etc. might be another idea that affects the outcomes; the extent to which the antibodies used in the various studies eliminate the possibility of residual expression of C-NAP1 might also be discussed.

The CEP250 antibody which was made in house with GST-C-NAP1¹⁹⁸⁴⁻²⁴⁴² is very sensitive (Jeong et al., JCS 120:2106, 2007). Since the antibody detects the C-terminal end of CEP250, it can detect most of the N-terminal truncated mutants. Immunoblot analysis revealed a prominent CEP250 band in the wild type RPE1 cells, but no specific band was detected in the *CEP250* KO cells, suggesting that smaller CEP250 proteins may not present in the *CEP250* KO cells (Attached Figure A2).

Attached Figure A2. Immunoblot analysis of CEP250 in the wild type and *CEP250* KO RPE-1 cells with the antibodies specific to CEP250 and GAPDH

\2. While Ryu et al. suggest in the Discussion that residual CROCC may affect the phenotype in their cells, Figure S2E seems to show a less marked loss of CROCC protein from the C-NAP1-deficient centrioles than was described by Panic et al. and by Flanagan et al. However, this experiment may have been done after serum starvation, so may not be directly comparable.

The question of how reduced the CROCC levels are in the currently-described C-NAP1 knockout should also be resolved, as this appears to be a significant difference from the previously-described effects of C-NAP1 loss on the centriolar linker.

Since both the reviewers pointed out centrosome localization of CROCC in the *CEP250* KO cells, we carefully examined this point in different cell density conditions. In cell density of 1.3×10^4 cells/cm², the cilia formation rate of RPE1 cells was about 60%, while that of the *CEP250* KO cells was about 40% (Fig. 2, B and F). When the cells were cultured in a high density, such as of 2.5×10^4 cells/cm², 80% of the wild type RPE1 had cilia (Figs. 2 F; S3 B). In the high cell density condition, the cilia formation rate of the *CEP250* KO cells increased to 75%, which is comparable to that of the wild type cells (Fig. 2 F). This result is consistent with the previous reports that cilia formation rates were not reduced in the *CEP250* KO cells (Panic et al., 2015; Mazo et al., 2016; Flanagan et al., 2017). We carefully predict that the other laboratories might perform cilia formation experiments at high densities and observed that 80% of the cells formed cilia in both the control and *CEP250* KO cells (Panic et al., 2015; Mazo et al., 2016; Flanagan et al., 2017).

We determined centrosome CROCC in the wild type and *CEP250* KO RPE1 cells. As expected, all the wild type cells had centrosome CROCC, irrespective of the presence or absence of cilia. In *CEP250* KO cells, almost of 90% of the cells with cilia include centrosome CROCC (Fig. 2 F). However, only about a half of the non-ciliated *CEP250* KO cells had CROCC at the centrosome area irrespective of the cell densities (Fig. 2 F). These results suggest that RPE1 cells in high density have more CROCC at the centrosomes and consequently cilia formation rates of the *CEP250* KO cells increased up to the wild type cell levels.

It remains to be determined why the centrosome CROCC increases in a high cell density condition. It is possible that, in a high cell density, mTORC1 might force the cells exit the cell cycle and the cells do not need to prepare for proliferation anymore. In such conditions, CROCC might be placed at the centrosomes via interactions with unknown centrosome components. In fact, our immunostaining results suggest that two centrioles are directly linked by the inter-centriolar/rootlet fibers even in the absence of CEP250 (Fig. 2 G) As far as CROCC is accumulated at the centrosomes, PCMI is accumulated near the centrosomes and cilia assembly rates increases. This point was described in the Discussion section of the revised manuscript.

3. Fig 1C appears to show an increase in CROCC over the serum starvation period, contrary to the authors' conclusion. A quantitation should be provided to clarify this issue.

As suggested, we repeated the experiments and quantified the bands. The mean numbers of the band intensities are indicated at Fig. 1 C of the revised manuscript.

4. The experiment shown in Fig S2D should be repeated and the relevant statistical information included.

As suggested, we repeated the experiments in Fig. S2 D and statistically analyzed the results.

5. The authors should describe the exonic position of the CRISPR cleavage sites used in the generation of the knockouts and the protein-level consequence of the disruptions. This is important to allow comparison with other work.

We targeted exon 1 for *CROCC* and *CEP250* KO and exon 3 *PCMI* KO. The KO lines were confirmed with immunostaining and immunoblot analyses (Figs. S1, S2 and S6). We described this point in the Materials and Methods section of the revised manuscript.

6. A uniform approach to labelling the statistical analyses should be taken throughout the paper (e.g., Fig. 3 uses * and 'n.s.', while Fig. 2 uses letters to indicate significance or otherwise). I suggest that the * and 'n.s.' is more conventionally used.

As suggested, we indicated the statistical significances between the important experimental groups with * and n.s..

7. The cellular volume examined to define the centrosome-proximal intensities of the satellite proteins should be specified.

We measured the centrosome intensities of the centriolar satellite proteins with circular area of $20 \mu\text{m}^2$ centered at the middle of the two centrioles. We described this point in the Materials and Methods section of the revised manuscript.

8. A rescue control should be shown for the *CEP72* siRNA ciliation experiment in Figure 4.

Unfortunately, we were not able to perform a rescue control experiment with a *CEP72* expression vector, due to limited time period of revision. It was very hard to manage cellular levels of ectopic FLAG-*CEP72* to the endogenous *CEP72* protein levels (data not shown). We realized that interpretable results might not be obtained unless the cellular levels of FLAG-*CEP72* are properly controlled to the endogenous levels.

9. The experiment shown in Fig. S3E is not explained clearly in the text and could be omitted as currently presented.

As suggested, we omitted Fig. S3 E from the revised manuscript

10. Details of the microscopy should be expanded to say whether images presented are from single sections or, if deconvolved images, how many sections were imaged; objective lens details should also be provided.

Most of the cells were observed with a conventional fluorescence microscope (Olympus; IX51) equipped with a CCD camera (Qimaging; Qicam Fast 1394) using PVCAM (version 3.9.0; Teledyne Photometrics). Images were saved as Adobe Photoshop 2021 (version 22.4.2). We also used super-resolution microscope (Carl Zeiss; ELYRA PS.1) for imaging *CROCC* (Figs. 1 F; 6 A) and confocal microscope (Carl Zeiss; LSM700) for Fig. 10 A. As suggested, we described how images were acquired in the materials and methods section in detail.

11. Additional work that addresses the roles of centriolar satellites (PCM1) in ciliogenesis should be considered, particularly in respect of (potential) cell type-specific aspects of the model: Hall et al. 2023 (PMID 36790165); Aydin et al. 2020 (PMID 32555591).

As suggested, we discussed tissue type-specific aspects of roles of centriolar satellites (PCM1) in ciliogenesis, citing both Hall et al. (2023) and Aydin et al. (2020) papers in the Discussion section.

Reviewer #2

The authors have addressed the majority of my comments and substantially improved the manuscript. I have a few minor points to make at this stage:

1) Fig 2G is not intuitive, especially the split boxes in the graph that show % of CROCC+ and CROCC- centrosomes in cells without cilia. According to the methods, groups with same letter are not significantly different but I don't see the point of comparing percentage of cells without cilia in CEP250 KO to percentage of cells with cilia in WT.

In the revised manuscript, we labeled the statistical significances with * and n.s., instead of letters. We indicated only meaningful statistical significance on the data so that the readers can easily understand the essence of the experiments.

We also expanded the Fig. 2 F to elucidate the discrepancy of our data with previous reports (Panic et al., 2015; Mazo et al., 2016; Flanagan et al., 2017), regarding cilia formation rates in CEP250 KO cells. In the revised manuscript, we cultured the CEP250 KO cells in a high density and determined the centrosome localization of CROCC. The results showed that the cilia formation rates increased when CEP250 KO cells were cultured in a high cell density (Fig. 2 F). At the same time, the proportion of the cells with centrosome CROCC increased (Fig. 2, F, G and H). These results suggest that the RPE1 cells became more stationary at G0 phase in a high-density culture, and, as a result, CROCC was better located at the centrosomes.

2) The degree of OFD1 reduction in basal bodies of CEP250KO and CROCC KO is surprising. While OFD1 associates with centriolar satellite, it has a large pool at distal appendages, where it is required for cilia formation. Given that CEP90 is still detectable, distal appendages probably still form in these cells, but the authors should highlight these points in the text.

As the reviewer pointed out, OFD1 is a centriole protein (Borgne et al., *Plos Biology*, 2022). CEP90 is also a centriole protein, even if both OFD1 and CEP90 are present is centriolar satellites (Kim et al., *Plos One*, 2012). We observed that the OFD1 and CEP90 signals were detected at the centrioles, even if they are absent from centriolar satellites of the CEP250 and CROCC KO cells (Fig. 3 A). Reduction in the centrosome signals of OFD1 and CEP90 are largely attributed to absence of centriolar satellites near the centrosomes (Fig. 3 B). We changed the OFD1 panels in Figure 3A for better representative ones.

3) In the discussion, it is stated "it is possible that CROCC directly associates with the proximal end of centrioles in the absence of CEP250". Being a rather important point, it would be straightforward to test this by high/super resolution microscopy.

As suggested, we presented better representative pictures of CROCC in the CEP250 KO cells (Fig. 2 G). We also carefully discussed this point in the Discussion section.

4) The authors state "All the truncated FLAG-PCM1 proteins were detected at the centrosomes except FLAG-PCM1 1201-2016 and FLAG-PCM1Δ551-1200, suggesting that the 551-1200 region of PCM1 is important for centrosome localization of PCM1" I agree but why is there no satellite staining of FL PCM1 in PCM1 KO cells? Can any of the truncated versions form satellites?

The full-length FLAG-PCM1 protein in Fig. 8 E does not look like centriolar satellites, largely due to inefficiency of the FLAG antibody. The FLAG-PCM1 protein looked like centriolar satellites when it was detected with the PCM antibody (Fig. S6 I).

5) The authors should highlight the caveat that the protein domains implicated in CROCC and PCM1 interaction are expansive and one cannot exclude other interactors contributing to the results.

The defined interaction regions of CROCC and PCM1 are about 550 residue-long, which are large. Therefore, we cannot rule out the possibility that other interacting proteins may contribute to the specific interactions between CROCC and PCM1. We described this point in the Discussion section of the revised manuscript.

December 11, 2023

RE: JCB Manuscript #202105065RR

Prof. Kunsoo Rhee
Seoul National University
Department of Biological Sciences
1 Gwanak-ro, Gwanak-gu
Seoul 08826
Korea, Republic of (South Korea)

Dear Prof. Rhee,

Thank you for submitting your revised manuscript entitled "The inter-centriolar fibers function as docking sites of centriolar satellites for cilia assembly." We would be happy to publish your paper in JCB pending final revisions necessary to meet our formatting guidelines (see details below).

A. MANUSCRIPT ORGANIZATION AND FORMATTING:

1) Text limits: Character count for Articles is < 40,000, not including spaces. Count includes title page, abstract, introduction, results, discussion, and acknowledgments. Count does not include materials and methods, figure legends, references, tables, or supplemental legends.

2) Figure formatting: Articles may have up to 10 main text figures. Please consolidate the model image to Figure 10 or move this into the supplement. Scale bars must be present on all microscopy images, including inset magnifications. Molecular weight or nucleic acid size markers must be included on all gel electrophoresis. Please add scale bars to magnifications in Figures 1D, 2A/D/G, 3A, 4B, 6B/D, 7A, 8A/E, & 10A.

Also, please avoid pairing red and green for images and graphs to ensure legibility for color-blind readers. If red and green are paired for images, please ensure that the particular red and green hues used in micrographs are distinctive with any of the colorblind types. If not, please modify colors accordingly or provide separate images of the individual channels.

3) Statistical analysis: Error bars on graphic representations of numerical data must be clearly described in the figure legend. The number of independent data points (n) represented in a graph must be indicated in the legend. Please, indicate whether 'n' refers to technical or biological replicates (i.e. number of analyzed cells, samples or animals, number of independent experiments). If independent experiments with multiple biological replicates have been performed, we recommend using distribution-reproducibility SuperPlots (please see Lord et al., JCB 2020) to better display the distribution of the entire dataset, and report statistics (such as means, error bars, and P values) that address the reproducibility of the findings.

Statistical methods should be explained in full in the materials and methods. For figures presenting pooled data the statistical measure should be defined in the figure legends. Please also be sure to indicate the statistical tests used in each of your experiments (both in the figure legend itself and in a separate methods section) as well as the parameters of the test (for example, if you ran a t-test, please indicate if it was one- or two-sided, etc.). Also, if you used parametric tests, please indicate if the data distribution was tested for normality (and if so, how). If not, you must state something to the effect that "Data distribution was assumed to be normal but this was not formally tested."

4) Materials and methods: Should be comprehensive and not simply reference a previous publication for details on how an experiment was performed. Please provide full descriptions (at least in brief) in the text for readers who may not have access to referenced manuscripts. The text should not refer to methods "...as previously described." Please also indicate the acquisition and quantification methods for immunoblotting/western blots. Please describe in full the immunoblotting methods including the type of membrane used and acquisition and quantification methods.

5) For all cell lines, vectors, constructs/cDNAs, etc. - all genetic material: please include database / vendor ID (e.g., Addgene, ATCC, etc.) or if unavailable, please briefly describe their basic genetic features, even if described in other published work or gifted to you by other investigators (and provide references where appropriate). Please be sure to provide the sequences for all of your oligos: primers, si/shRNA, RNAi, gRNAs, etc. in the materials and methods. You must also indicate in the methods the

source, species, and catalog numbers/vendor identifiers (where appropriate) for all of your antibodies, including secondary. If antibodies are not commercial, please add a reference citation if possible.

6) Microscope image acquisition: The following information must be provided about the acquisition and processing of images:

- Make and model of microscope
- Type, magnification, and numerical aperture of the objective lenses
- Temperature
- Imaging medium
- Fluorochromes
- Camera make and model
- Acquisition software
- Any software used for image processing subsequent to data acquisition. Please include details and types of operations involved (e.g., type of deconvolution, 3D reconstitutions, surface or volume rendering, gamma adjustments, etc.).

7) References: There is no limit to the number of references cited in a manuscript. References should be cited parenthetically in the text by author and year of publication. Abbreviate the names of journals according to PubMed.

8) Supplemental materials: Articles are generally allowed up to 5 supplemental figures and 10 videos. You currently exceed this limit but, in this case, we will be able to give you the extra space. Please also note that tables, like figures, should be provided as individual, editable files. A summary of all supplemental material should appear at the end of the Materials and methods section. Please include one brief sentence per item.

9) eTOC summary: A ~40-50 word summary that describes the context and significance of the findings for a general readership should be included on the title page. The statement should be written in the present tense and refer to the work in the third person. It should begin with "First author name(s) et al..." to match our preferred style.

10) Conflict of interest statement: JCB requires inclusion of a statement in the acknowledgements regarding competing financial interests. If no competing financial interests exist, please include the following statement: "The authors declare no competing financial interests." If competing interests are declared, please follow your statement of these competing interests with the following statement: "The authors declare no further competing financial interests."

11) A separate author contribution section is required following the Acknowledgments in all research manuscripts. All authors should be mentioned and designated by their first and middle initials and full surnames. We encourage use of the CRediT nomenclature (<https://casrai.org/credit/>).

12) ORCID IDs: ORCID IDs are unique identifiers allowing researchers to create a record of their various scholarly contributions in a single place. Please note that ORCID IDs are required for all authors. At resubmission of your final files, please be sure to provide your ORCID ID and those of all co-authors.

13) JCB now requires authors to submit Source Data used to generate figures containing gels and Western blots with all revised manuscripts. This Source Data consists of fully uncropped and unprocessed images for each gel/blot displayed in the main and supplemental figures. Since your paper includes cropped gel and/or blot images, please be sure to provide one Source Data file for each figure that contains gels and/or blots along with your revised manuscript files. File names for Source Data figures should be alphanumeric without any spaces or special characters (i.e., SourceDataF#, where F# refers to the associated main figure number or SourceDataFS# for those associated with Supplementary figures). The lanes of the gels/blots should be labeled as they are in the associated figure, the place where cropping was applied should be marked (with a box), and molecular weight/size standards should be labeled wherever possible. Source Data files will be directly linked to specific figures in the published article.

14) Journal of Cell Biology now requires a data availability statement for all research article submissions. These statements will be published in the article directly above the Acknowledgments. The statement should address all data underlying the research presented in the manuscript. Please visit the JCB instructions for authors for guidelines and examples of statements at (<https://rupress.org/jcb/pages/editorial-policies#data-availability-statement>).

B. FINAL FILES:

Thank you for this interesting contribution, we look forward to publishing your paper in Journal of Cell Biology.

Sincerely,

Karen Oegema, PhD
Monitoring Editor
Journal of Cell Biology

Dan Simon, PhD
Scientific Editor
Journal of Cell Biology